# GFI1 facilitates efficient DNA repair by regulating PRMT1 dependent methylation of MRE11 and 53BP1

Charles Vadnais[1], Riyan Chen[1], Jennifer Fraszczak[1], Zhenbao Yu[2], Jonathan Boulais[1], Jordan Pinder[3], Daria Frank[4], Cyrus Khandanpour [4], Josée Hébert[5,6,7], Graham Dellaire[3], Jean-François Côté[1,8], Stéphane Richard[2,9,10,11], Alexandre Orthwein[2,10,11,12], Elliot Drobetsky[8] & Tarik Möröy[1,11,13]

GFI1 is a transcriptional regulator expressed in lymphoid cells, and an "oncorequisite" factor required for development and maintenance of T-lymphoid leukemia. GFI1 deletion causes hypersensitivity to ionizing radiation, for which the molecular mechanism remains unknown. Here, we demonstrate that GFI1 is required in T cells for the regulation of key DNA damage signaling and repair proteins. Specifically, GFI1 interacts with the arginine methyltransferase PRMT1 and its substrates MRE11 and 53BP1. We demonstrate that GFI1 enables PRMT1 to bind and methylate MRE11 and 53BP1, which is necessary for their function in the DNA damage response. Thus, our results provide evidence that GFI1 can adopt non-transcriptional roles, mediating the post-translational modification of proteins involved in DNA repair. These findings have direct implications for treatment responses in tumors overexpressing GFI1 and suggest that GFI1's activity may be a therapeutic target in these malignancies.

[1] Institut de Recherches Cliniques de Montréal, IRCM, Montréal, QC H2W 1R7, Canada. [2] Segal Cancer Centre, Lady Davis Institute for Medical Research, Jewish General Hospital, Montreal, QC H3T 1E2, Canada. [3] Departments of Pathology and Biochemistry & Molecular Biology, Dalhousie University, Halifax, NS B3H 4R2, Canada. [4] Department of Hematology, University Hospital, Essen 45147, Germany. [5] Department of Medicine, Université de Montréal, Montreal H3T 1J4 QC, Canada. [6] Division of Hematology-Oncology, Maisonneuve-Rosemont Hospital, Montreal H1T 2M4 QC, Canada. [7] Quebec Leukemia Cell Bank, Maisonneuve-Rosemont Hospital, Montreal H1T 2M4 QC, Canada. [8] Département de Médecine, Université de Montréal and Centre de Recherche, Hôpital Maisonneuve Rosemont, Montréal, QC H1T 2M4, Canada. [9] Deparment of Medicine, McGill University, Montreal H4A 3J1 QC, Canada. [10] Department of Oncology, McGill University, Montreal, QC H4A 3T2, Canada. [11] Division of Experimental Medicine, McGill University, Montreal, QC H3A 1A3, Canada. [12] Department of Microbiology and Immunology, McGill University, Montreal, QC H3A 2B4, Canada. [13] Département de Microbiologie, Infectiologie et Immunologie, Université de Montréal, Montréal, QC H3C 3J7, Canada. Correspondence and requests for materials should be addressed to T.M. (email: Tarik.Moroy@ircm.qc.ca)

The GFI1 protein is primarily known as a transcription factor essential for hematopoiesis and, in particular, controls the differentiation of myeloid and lymphoid cells from hematopoietic stem and precursor cells. During early hematopoiesis, GFI1 represses critical target genes in bi-potential or multi-potential cells thereby affecting their lineage commitment. It exerts this effect by recruiting the histone de-methylase LSD1 and histone de-acetylases, including HDAC1 to down-regulate promoter activity[1]. In addition to its function in hematopoietic differentiation, GFI1 is involved in regulating cell survival. Early studies showed that GFI1 exhibits anti-apoptotic properties upon overexpression in T cells[2,3]. Consistent with this, we recently demonstrated that GFI1-deficient T cells exhibit increased sensitivity to ionizing radiation (IR), which induces highly lethal DNA double-strand breaks (DSB), suggesting a role for GFI1 in the DNA damage response (DDR) through a yet unknown mechanism[4].

Following induction of DSBs, cells elicit a complex response including two major DNA repair pathways: (i) non-homologous end joining (NHEJ) where DSBs are directly ligated, and which can take place throughout the cell cycle[5–7] and (ii) homologous recombination (HR), which requires a homologous DNA template thereby occurring exclusively in the S and G2 phases[5]. The cellular response to DSBs leading to HR is triggered via recruitment of the trimeric MRN complex, composed of the proteins MRE11, RAD50, and NBS1, to sites of damage. This complex mediates recruitment of the ataxia telangiectasia mutated (ATM) serine/threonine kinase, which becomes activated by monomerization and auto-phosphorylation[5,8,9]. ATM initiates signaling from DSBs by phosphorylating numerous downstream targets, including the histone variant H2AX to form γ-H2AX[10,11]. Activation of the closely related kinase ataxia telangiectasia and Rad3-related (ATR) is thought to occur later on during the DDR in response to replication protein-A- (RPA-) coated stretches of single-stranded DNA (ssDNA)[5,12–14]. Such ssDNA can be generated at stalled replication forks or during resection of DSBs via a combination of MRE11 and EXO1/BLM nuclease activities[5,15,16].

The ATM/ATR protein phosphorylation cascade is complemented by additional post-translational modifications (PTMs) that regulate cellular responses to genotoxic stress. Protein arginine methyltransferase 1 (PRMT1) methylates a number of DDR targets and abrogation of its activity causes hypersensitivity to DNA damage, defects in cell cycle control, and an accumulation of chromosomal abnormalities[17]. Of particular interest here, PRMT1 targets MRE11 as well as 53BP1, both of which are critical for DNA repair pathway choice: MRE11 by initiating DNA end resection thus promoting HR, and 53BP1 by inhibiting inappropriate resection of DNA ends during G1 to favor NHEJ[16,18].

MRE11 contains a glycine- and arginine-rich sequence termed the GAR motif. Methylation of this motif by PRMT1 is required for the processive exonuclease activity of MRE11 during end resection, and for S phase checkpoint control, but not for its interaction with other members of the MRN complex[19,20]. Importantly, cells expressing a non-methylable mutant MRE11 with arginine to lysine (R/K) substitutions within the GAR motif display increased sensitivity to IR, reduced focus formation of the HR marker RAD51[21], ATR activation defects, and genomic instability[19].

53BP1 also contains a GAR motif that is methylated by PRMT1. This motif is essential for 53BP1's localization to sites of damage and its methylation is required for 53BP1's DNA binding capacity[22], but not for its oligomerization[23]. PRMT1 has also been shown to methylate BRCA1, hnRNPK and hnRNPUL1, all of which are known to play some role in the DDR[24–27].

Here we describe a previously unknown, non-transcriptional role for GFI1 as a mediator of post-translational modifications of key DNA repair proteins. Our data indicate that, in T cells, GFI1 is required for the interaction of PRMT1 with MRE11 and 53BP1, and for their subsequent methylation. Moreover, in cells lacking GFI1, both MRE11 and 53BP1 remain hypo-methylated and DNA repair is compromised. These findings may have direct implications for GFI1 as a therapeutic target in malignancies where GFI1 is involved such as T-cell leukemia, neuroendocrine lung carcinomas[28], and in medulloblastoma, where it is believed to be a driving oncogene in certain aggressive subgroups[29].

## Results

**GFI1 promotes cell survival following DNA damage**. To assess the role of GFI1 in the DDR, we exploited an established *Gfi1* knock-out mouse model[30]. In addition, to evaluate potential dose-dependent effects of Gfi1, we used a previously described *Gfi1* knock-in (KI) strain expressing the human *GFI1* gene in place of the murine counterpart, and a knockdown (KD) strain expressing reduced levels of the "knocked in" human *GFI1*[31,32] (Supplementary Figure 1a). We also employed two cultured human cell models: (i) SupT1 T-cell lymphoblasts ectopically overexpressing GFI1 to determine the effects of increased GFI1 expression (Supplementary Fig. 1b) and (ii) a Jurkat T-cell leukemia line in which *GFI1* was knocked-out using a Crispr/Cas9 strategy (Supplementary Fig. 1c).

Annexin V staining showed that *Gfi1* KO thymocytes display higher levels of apoptosis, either at baseline or following exposure to IR, compared to WT controls (Supplementary Fig. 1d), consistent with previous results in T-cell lymphoma and T-cell acute lymphoblastic leukemia (T-ALL)[4]. Cells from Gfi1 KD mice showed a gene dosage effect, since they displayed both higher baseline and post-IR levels of apoptosis than *Gfi1* KI control cells, but a less pronounced phenotype than *Gfi1* KO cells (Supplementary Fig. 1e). We also measured the changes in the number of live cells in populations of GFI1-overexpressing SupT1 T lymphoblasts vs. vector controls following exposure to DNA damaging agents, and found that the former were able to better recover and proliferate following exposure to different doses of IR or of the nucleoside analog cytarabine (Fig. 1a, b). Cytarabine is one of the most commonly used genotoxic chemotherapeutic for leukemia and lymphoma and kills cells by becoming incorporated into replicating DNA, leading to replication fork stalling and collapse. In contrast to the situation for SupT1 cells, Jurkat T-ALL *GFI1* knock-out cells were less able to recover from exposure to IR or cytarabine compared to parental counterparts (Fig. 1c, d). Notably, in both models, there was no difference in proliferation of untreated cells expressing different levels of GFI1. (Supplementary Fig. 1f-h). These results validated the use of engineered Supt1 and Jurkat cell lines for investigating the role of GFI1 in the DDR since they are in agreement with earlier results showing the involvement of GFI1 in the regulation of cell survival after DNA damage[4].

**GFI1 activity promotes repair of DNA strand breaks**. To assess a potential role for GFI1 in DNA repair after IR exposure, we performed comet assays under alkaline conditions, which quantify removal of DNA breaks but do not distinguish between DSBs and single strand breaks (SSBs). We observed that GFI1 over-expressing SupT1 cells repair DNA breaks more rapidly than control cells (Fig. 1e). Similarly, thymocytes from *Gfi1* KO mice repaired DNA breaks more slowly than thymocytes from WT control mice (Fig. 1f). Interestingly, *Gfi1* KD thymocytes removed DNA breaks more slowly than *Gfi1* KI cells, but faster than *Gfi1* KO cells, indicating a dose-dependent effect of GFI1 on DNA

repair (Fig. 1g). These observations were supported in the Jurkat KO model, where GFI1-deficient cells displayed less efficient repair vs. the parental control (Fig. 1h). Importantly, the deficiency in DNA repair was also observed in GFI1 KO Jurkat cells using a comet assay under neutral conditions, which measures the repair of DSBs exclusively (Supplementary Fig. 1i). Additionally, comet assays comparing DNA repair efficiency between thymocytes extracted from *Gfi1* KO *p53* KO mice and *Gfi1* wt *p53* KO

mice showed the same repair defect in the GFI1 KO cells as seen in the p53 WT context (Supplementary Fig. 1j), showing that the effect of GFI1 on DNA repair is independent of p53 status. Finally, to ensure that our results are not attributable to differential cell cycle-dependent effects on repair, we verified that cells expressing different levels of GFI1 had a similar proportion of cells in each phase of the growth cycle (Supplementary Fig. 2a, b).

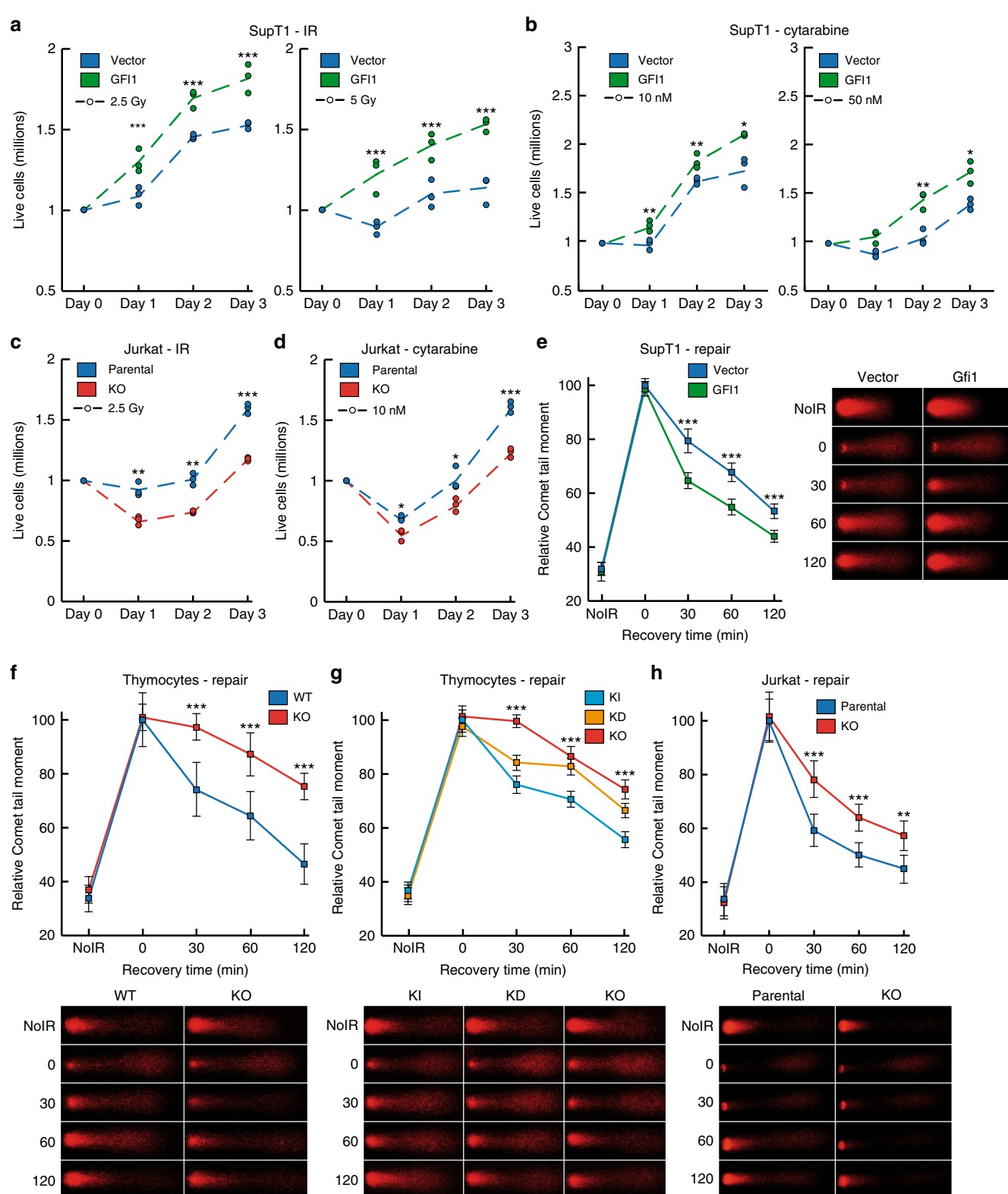

In order to probe the effect of GFI1 on DNA repair specifically of DSBs, we first measured levels of γ-H2AX in response to IR by immunofluorescence. As expected, following exposure to IR, thymocytes expressing WT levels of GFI1 protein showed an increase in the intensity of nuclear γ-H2AX signal, which then receded over time (suggesting resolution of DSBs). By comparison, Gfi1 KO cells showed a delayed increase in nuclear γ-H2AX, moreover the signal took longer to recede after reaching a maximum (Fig. 2a). The γ-H2AX signal similarly persisted longer in GFI1 KD cells compared to Gfi1 KI (Supplementary Fig. 2c). Consistently, γ-H2AX peaked at lower levels and the signal was less persistent in SupT1 cells overexpressing GFI1 compared to empty vector counterparts, again suggesting faster resolution of DSBs in cells expressing more GFI1 protein (Fig. 2b). Measurements of γ-H2AX signal by FACS showed similarly higher and more persistent levels in GFI1 KO thymocytes compared to WT cells, as well as in vector control SupT1 cells compared to GFI1 overexpressing counterparts (Fig. 2c, d). Interestingly, the increase in γ-H2AX levels and persistence in GFI1 KO thymocytes was present in cells regardless of the phase of the cell cycle they were in (Supplementary Fig. 2d).

Next, we measured the efficiency of DSB repair by HR in WT vs. GFI1-deficient Jurkat cells more directly by electroporating the cells with a vector expressing Cas9 and one of two different guide RNAs for the Lamin A locus along with a template plasmid that, when recombined with the genomic target locus expresses the Clover green fluorescent protein[33]. The results with both gRNAs indicate that the efficiency of HR repair was markedly reduced in GFI1 KO cells (Fig. 2e). We also measured NHEJ by electroporating WT vs. GFI1-deficient Jurkat cells with the EJ5-GFP plasmid, which contains a silent GFP coding cassette whose expression can be restored following NHEJ-mediated repair of I-SceI-induced breaks[34]. We did not find a significant decrease in the efficiency of NHEJ in Gfi1 KO vs. parental cells (Supplementary Fig. 2e).

Given the above results revealing a role for Gfi1 in DSB repair by HR, we hypothesized that abrogation of GFI1 may render cells more susceptible to chromosomal instability. However, chromosomal breakage studies in metaphases of anti-CD3 stimulated peripheral T cells extracted from GFI1 WT vs. KO mice did not show significant chromosomal instability in the absence of GFI1 (Supplementary Fig. 2f). Even when carrying out this analysis in a p53 deficient background to avoid loss of damaged cells due to p53-mediated apoptosis, no significant increase in the number of metaphase breaks in GFI1 KO cells compared to WT counterparts was observed (Supplementary Fig. 2g, h). This indicates that while Gfi1 is required for survival and DSB repair during genotoxic stress, its absence does not necessarily engender chromosomal instability.

**GFI1 interacts with the DDR proteins MRE11 and PRMT1.** To gain more insight into how GFI1 influences DNA repair, we set out to identify its potential binding partners by AP-MS (Affinity purification mass spectrometry) analysis of proteins that co-immunoprecipitate with a GFI1-Flag fusion protein from 293 T cells. This approach identified several known GFI1 interacting proteins such as LSD1, HDACs and CoRest, in addition to candidates that play well-defined roles in the recognition and repair of DSBs, notably members of the MRN complex Mre11 and Rad50 (Fig. 3a–c, Table 1, Supplementary Data 1). The PRMT1 methyltransferase, which post-translationally modifies MRE11[20], was also identified in this experiment. Independent co-immunoprecipitations confirmed Flag-GFI1 interactions between with MRE11 and PRMT1 (Fig. 3d, lane 1). Co-immunoprecipitation with truncated forms of GFI1 showed that its intermediate domain is required for interaction with these DDR-related proteins (Fig. 3d, lanes 2–4, Fig. 3e, Supplementary Fig. 4a-c), in contrast to its interaction with histone modifying enzymes such as LSD1, which rely on the N-terminal SNAG domain of GFI1[35]. Interestingly, deletion of DNA binding zinc finger domain of GFI1 did not affect interaction with PRMT1 or MRE11 and consistently, the interaction between the full GFI1 protein and MRE11 was independent of DNA binding, as shown by co-IP experiments carried out in the presence of benzonase (Fig. 3f).

We tested whether the DNA repair defects observed in the absence of Gfi1 were due to the interaction between Gfi1 and MRE11 or PRMT1 by transfecting GFI1 KO Jurkat cells with plasmids expressing the different GFI1 constructs depicted in Fig. 3e. We found that while full length GFI1 protein rescued the ability of Gfi1 KO cells to repair DNA damage as measured by comet assays, GFI1 protein with a deletion of the intermediate domain did not. In addition, GFI1 proteins with either a deletion of the SNAG domain or of the DNA-binding zinc finger domain also rescued the DNA repair defect in GFI1-deficient cells (Fig. 3g). This is consistent with the ability of these GFI1 mutants to interact with PRMT1 and MRE11 (Fig. 3d). We further validated these findings by showing an interaction between GFI1 and both PRMT1 and MRE11 in non-transfected Jurkat and SupT1 cells indicating that GFI1 can bind to both proteins at endogenous expression levels (Fig. 3h,i). These results confirm that the interactions of GFI1, through its intermediary domain, with PRMT1 and MRE11 is required for its activity in DNA repair, but suggest that GFI1's interaction with DNA is not.

Interestingly, this analysis also led to the identification of the ATM protein with a single peptide. While preys identified by a single peptide are generally not considered significant, we were able to co-immunoprecipitate ATM with the same set of GFI1 fusion proteins as MRE11 and PRMT1 (Fig. 3e). We assessed whether GFI1 deficiency had an effect on ATM activity following IR exposure by immunofluorescence, but found that the appearance of p-ATM foci did not differ between GFI1 KO and GFI1 wt cells (Supplementary Fig. 3, 4d), suggesting that the effect of GFI1 on DNA repair is not mediated through ATM.

**GFI1 activities are independent of DNA damage.** We next aimed to determine if the interaction between GFI1 and DDR proteins would be modulated by genotoxic stress. Also, although

---

**Fig. 1** GFI1 promotes survival and repair of DNA damage. **a** GFI1 overexpressing SupT1 cells and vector control cells were seeded at 1 million cells per ml and exposed to 2.5 (left panel) or 5 (right panel) Gy IR. Cells were counted each following day. Dashed lines show average cell numbers and individual data points of a triplicate experiment are shown. *p < 0.05, **p < 0.01, ***p < 0.001 on a Welch corrected T-test. **b** SupT1 cells as in A were exposed to 10 (left panel) or 50 (right panel) nM Cytarabine. Cells were counted and results are presented as in **a**. **c** Gfi1 KO Jurkat T cells and parental control cells were seeded at 1 million cells per ml and exposed to 2.5 Gy IR. Cells were counted and results are presented as in **a**. **d** Jurkat cells as in **c** were exposed to 10 nM Cytarabine. Cells were counted and results are presented as in **a**. **e** GFI1 overexpressing SupT1 cells and vector controls were exposed to 5 Gy and allowed to recover for the indicated time. Cells were then lysed and analyzed by alkaline Comet assay. Comet tail moment averages are shown with representative images on the right. One of three replicate experiments is shown. Error bars represent s.d. **f** Thymocytes extracted from Gfi1 WT and age and sex matched Gfi1 KO mice were treated and analyzed as in **e**. Representative images are shown below the graph. **g** Thymocytes from Gfi1 KI, KD and KO mice were treated and analyzed as in **e**. **h** Jurkat cells with GFI1 KO and parental control cells were treated and analyzed as in **e**

the interaction of GFI1 with DNA did not appear to be required for its activity, we reasoned that GFI1 might still localize to sites of DNA damage indirectly through its interaction with DDR proteins and tested whether this was the case. However, co-IPs performed at different time points post-IR showed that the interaction between GFI1 and MRE11 or ATM was not affected by exposure to IR (Fig. 4a). In addition, immunofluorescence

experiments indicated no change in the nuclear distribution of endogenous GFI1 post-IR, and no co-localization with γ-H2AX (Fig. 4b).

However, since detection of endogenous proteins at sites of DNA damage can be challenging, we evaluated localization of a Gfi1-GFP fusion protein. Standard immunofluorescence did not reveal any co-localization of the fusion protein with γ-H2AX

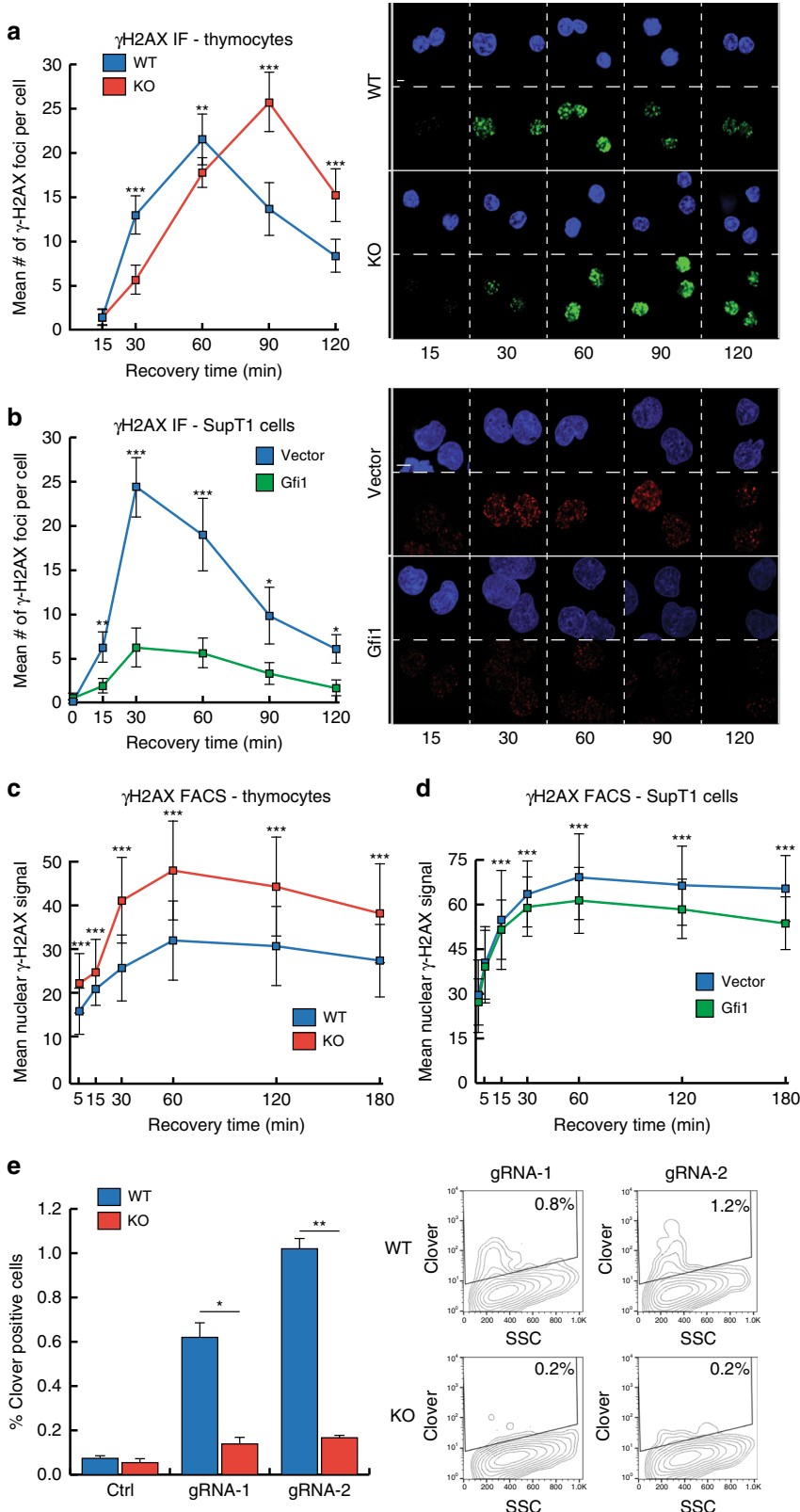

following IR exposure (Supplementary Fig. 4d). Immunofluorescence experiments in cells carrying an integrated LacO array and expressing both a LacR-Fok1-mCherry fusion endonuclease and the GFI1-GFP fusion protein showed no localization of the GFI1-GFP fusion protein to the site of Fok1 nuclease activity, to which γ-H2AX readily localized (Fig. 4c). Finally, experiments using UV laser micro-irradiation[36,37] of cells expressing GFI1-GFP did not show any localization of the fusion protein to sites of damage compared to GFP protein alone, whereas an mRuby-Ku80 fusion protein was recruited as expected[36] (Fig. 4d). We conclude from these experiments that the physical interaction of GFI1 with the DNA repair machinery and its effect on the cellular DNA repair capacity precedes the occurrence of DNA damage and does not involve localization of GFI1 to sites of DNA damage.

**GFI1 mediates methylation of MRE11 and 53BP1 by PRMT1**. Given that DNA damage did not appear to affect interactions or localization of GFI1, we hypothesized that GFI1 may mediate regulatory events that precede the activation of the DDR machinery and as such examined the potential role of GFI1 in mediating asymmetrical dimethylation of arginine residues by PRMT1, since PRMT1 is known to methylate a number of DDR proteins including MRE11 which is recruited to DSB sites during HR repair.

Co-immunoprecipitation experiments showed that the presence of asymmetric dimethylarginine (ADMA) on MRE11 is severely reduced in *Gfi1* KO thymocytes (Fig. 5a). Conversely, ADMA on MRE11 was increased in SupT1 cells overexpressing GFI1 (Fig. 5b). Importantly, the interaction between PRMT1 and MRE11 was absent in *Gfi1* KO thymocytes compared to WT cells, despite both proteins still being normally expressed (Fig. 5c, d) and was conversely increased in GFI1 overexpressing SupT1 cells compared to vector control cells. (Fig. 5e, f). This prompted us to investigate whether GFI1 might modulate the interaction between PRMT1 and other known DDR proteins, thereby affecting their methylation status. Interestingly, ADMA on 53BP1 was also reduced in *Gfi1* KO thymocytes compared to WT cells (Supplementary Fig. 5a) and, here again, the interaction between PRMT1 and 53BP1 was significantly reduced in the absence of GFI1 (Supplementary Fig 5b, c). Notably, while 53BP1 was not identified as a putative GFI1 binding partner in the mass spectrometry experiment, we were able to detect an interaction between endogenous GFI1 and 53BP1 proteins by co-immunoprecipitation (Supplementary Fig. 5d).

While we show that ADMA on MRE11 and 53BP1 is affected by GFI1, we could however confirm that the overall ADMA pattern of cell extracts from *Gfi1* KO thymocytes and from GFI1 overexpressing SupT1 cells was not significantly altered compared to their respective controls (Fig. 5g, h). Furthermore, in vitro methylation experiments using purified PRMT1, MRE11-GAR substrate and increasing concentrations of GFI1 protein showed no effect of the latter on the catalytic activity of PRMT1 (Supplementary Fig. 6). These results support the notion that GFI1 mediates methylation of a specific set of proteins by PRMT1

including MRE11 and p53BP1. In further support of this, a decrease in ADMA on MRE11 and 53BP1, a reduced interaction of both proteins with PRMT1 as well as maintenance of the overall ADMA pattern was also observed in GFI1-deficient Jurkat cells (Supplementary Fig. 7a-d). Changes in expression of GFI1 had no effect on immunoprecipitation efficiency of any of the proteins of interest (Supplementary Fig. 8a-f).

These results indicate that GFI1 is required for proper interaction between PRMT1 and both MRE11 and 53BP1, and efficient arginine dimethylation of the latter two proteins.

However, immunofluorescence experiments examining the localization of MRE11 and 53BP1 following exposure of cells to IR showed no change in the localization pattern of either protein in GFI1 KO cells compared to wt cells (Supplementary Fig. 9a, b), which is in agreement with prior findings that the methylation of these proteins is not required for their localization to sites of damage, but rather for their activity at sites of damage[19,22].

Additionally, we performed co-immunoprecipitation experiments of the ATM protein and while we found an interaction between ATM and MRE11, as has been reported in the literature[5,8,9], we found no interaction between ATM and PRMT1 (Supplementary Fig. 9c, d) suggesting that ATM is not related to this specific mechanism of regulation of MRE11 activity and is unlikely to mediate GFI1's effect on DNA repair efficiency.

In order to gain further support for the notion that part of the mechanism of action of GFI1 is through methylation of the GAR motif of MRE11, we used Mouse Embryonic Fibroblast (MEF) cells extracted from mice carrying R/K substitutions in MRE11, which cannot be methylated by PRMT1 (Fig. 6a) to assess whether they displayed phenotypes consistent with those observed in GFI1 KO cells. Indeed, we observed that cells expressing MRE11 R/K presented a delay in the repair of DNA breaks as measured by Comet assays (Fig. 6b), which is consistent with our hypothesis.

We hypothesized further that if GFI1 has an important mechanistic effect on PRMT1 activity, treatment of cells with an inhibitor of PRMT1 should eliminate any differences in cell survival or DNA repair between cells expressing different levels of GFI1. To test this, we exposed cells to MS023, a methyltransferase inhibitor selectively targeting class I PRMTs, including PRMT1[38], as there are currently no PRMT1 specific inhibitors available. Treatment with the inhibitor decreased the levels of ADMA on MRE11 (Fig. 6c) and caused a reduction of global cellular ADMA without having any apparent effect on cell viability (Supplementary Fig. 10). We then found that the difference in recovery and proliferation following IR exposure previously seen between GFI1 WT and KO Jurkat cells was eliminated by pre-treatment with the inhibitor, with both treated lines recovering at the same rate as the untreated GFI1 KO line (Fig. 6d, left). Similarly, pre-treating the cells with the inhibitor eliminated the difference in recovery after IR between GFI1 overexpressing SupT1 cells and vector controls. Treated cells from both lines recovered less efficiently than untreated vector control cells, which we believe reflects the decrease in PRMT1 activity to below its baseline level (Fig. 6d, right).

---

**Fig. 2** GFI1 affects γ-H2AX signaling and HR following IR. **a** Thymocytes extracted from *Gfi1* WT and age and sex matched *Gfi1* KO mice were exposed to 5 Gy IR and allowed to recover for the indicated time. Cells were spread on glass slides using a cytospin, fixed and stained for γ-H2AX. The mean number of γ-H2AX foci are shown with representative images on the right. One of three replicate experiments is shown. Error bars represent s.d. *$p < 0.05$, **$p < 0.01$, ***$p < 0.001$ on a Welch corrected *T*-test. Scale bar represents 10 μm. **b** GFI1 overexpressing SupT1 cells and vector controls were treated as in **a**. **c** Thymocytes extracted as in **a** were stained for γ-H2AX and analyzed by FACS. Mean γ-H2AX signal is shown. **d** GFI1 overexpressing SupT1 cells and vector controls were treated as in **c**. **e** Left: Jurkat cells with GFI1 KO and parental control cells were electroporated with plasmids encoding Cas9 and one of two gRNAs targeting the Lamin A locus and a donor plasmid. Cells electroporated without gRNA are shown as control. Right: Representative FACS profile showing Clover signal vs. side scatter. Positive gate determined using cells electroporated without gRNA plasmid

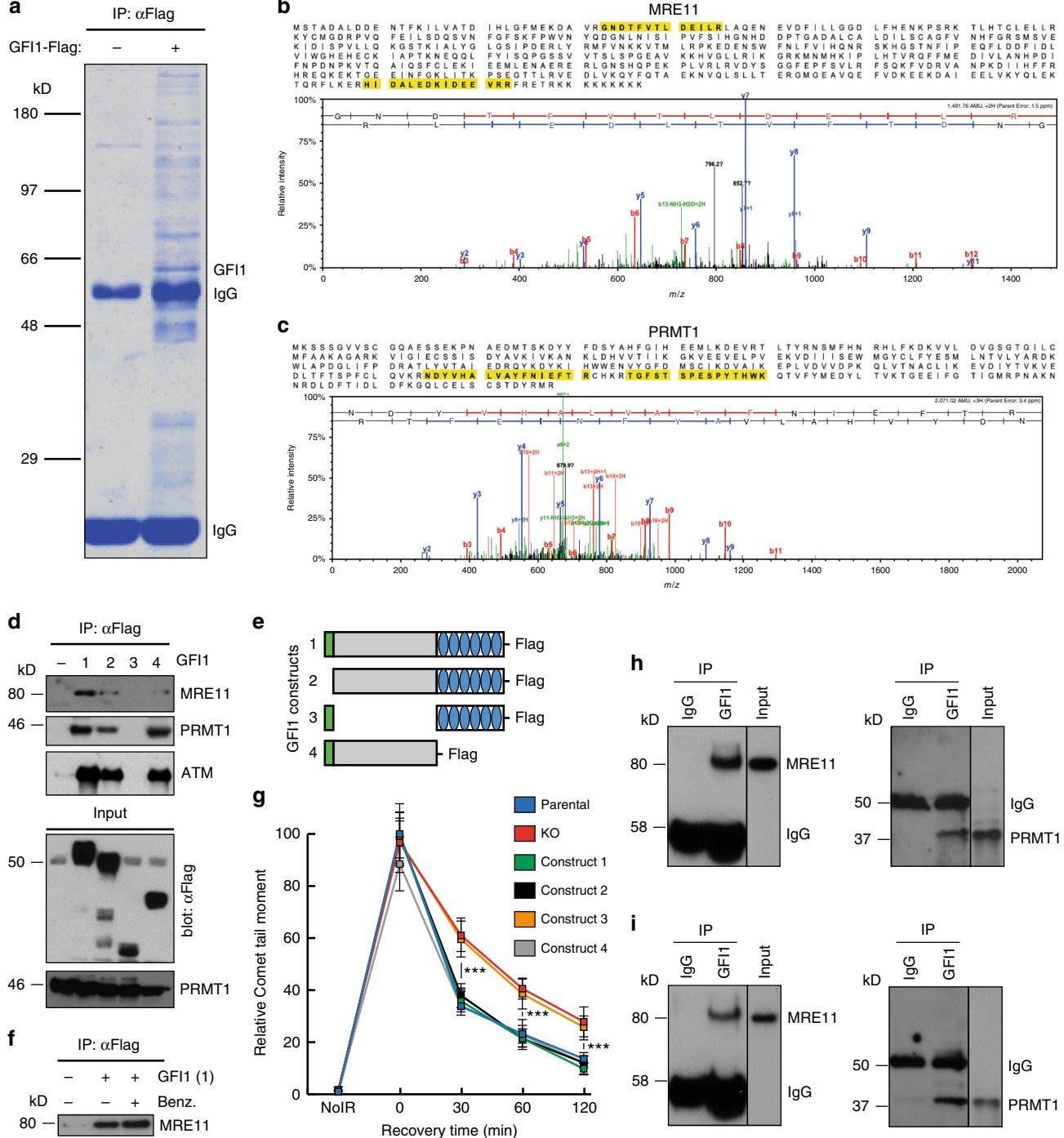

**Fig. 3** GFI1 interacts with proteins involved in the DDR. **a** GFI1-Flag fusion protein was immunoprecipitated from 293T cells. Co-precipitated proteins were run on polyacrylamide gel and stained with Coomassie blue. **b** Peptide sequence of MRE11 with peptides identified through mass spectrometry highlighted and corresponding spectra below. **c** Peptide sequence of PRMT1 with peptides identified through mass spectrometry highlighted and corresponding spectra below. **d** Variants of the GFI1-Flag fusion protein were immunoprecipitated from 293T cells. Extracts were separated by SDS–PAGE and blotted for the indicated proteins. **e** Diagram of the different GFI1 fusion proteins used in **d** and **g**. **f** GFI1-Flag fusion protein was immunoprecipitated in 293T cells in the presence or absence of benzonase. Extracts were separated by SDS–PAGE and blotted for MRE11. **g** GFI1 KO Jurkat cells were electroporated with plasmids expressing GFI1 variant constructs as shown in **e**. GFI1 KO and parental control Jurkat cells were used as controls. After 24 h, cells were exposed to 5 Gy IR and allowed to recover for the indicated time. Cells were lysed and analyzed by alkaline Comet assay. Comet tail moment averages are shown. One of three replicate experiments is shown. Error bars represent s.d. **h** Endogenous GFI1 protein was immunoprecipitated in SupT1 cells. Co-precipitated proteins were separated by SDS–PAGE and blotted for MRE11 and PRMT1. **i** Endogenous GFI1 protein was immunoprecipitated in Jurkat cells. Co-precipitated proteins were separated by SDS–PAGE and blotted for MRE11 and PRMT1

**Table 1 Selected Gfi1 binding partners**

| Gene symbols | Fold enrichment | BFDR |
|---|---|---|
| LSD1 | 15.39 | 0.00 |
| CoRest1 | 39.04 | 0.00 |
| CoRest3 | 10.28 | 0.00 |
| HDAC1 | 3.42 | 0.00 |
| HDAC2 | 2.21 | 0.00 |
| *DDR-related Proteins* | | |
| DNA-PK | 3.78 | 0.00 |
| MRE11 | 1.81 | 0.00 |
| PRMT1 | 2.21 | 0.00 |
| RAD50 | 2.21 | 0.00 |

Selected proteins identified by Mass-Spec as Gfi1 binding partners with their calculated fold enrichment and the Bayesian False Discovery Rate (BFDR)

In addition, the differences in repair of DNA breaks after IR which we had observed in Comet assays between GFI1 WT and KO Jurkat cells were also eliminated by MS023 treatment (Fig. 6e, left). And similarly, the repair efficiencies of MS023 treated GFI1 overexpressing SupT1 cells and vector controls were equally reduced to below that of untreated vector control cells (Fig. 6e, right).

To confirm that the role of GFI1 is mediated specifically through PRMT1, we performed siRNA mediated knockdown of PRMT1 in GFI1 overexpressing and vector control SupT1 cells (Supplementary Fig. 11a) and found by Comet assay that the knockdown reduced the DNA repair capacity of both cells lines to an equal level below that of the non-treated vector control cells (Fig. 6f), corroborating the results obtained with the MS023 inhibitor. In addition, siRNA knockdown of PRMT1 in GFI1 KO and WT Jurkat cells (Supplementary Fig. 11b) caused a decrease in the DNA repair capacity of GFI1 WT cells, but not in the GFI1 KO cells (Fig. 6g).

Finally, overexpression of PRMT1 in parental Jurkat cells (Supplementary Fig. 11c) lead to an improvement in the DNA repair capacity of the cells as measured by comet assay, whereas PRMT1 overexpression in GFI1 KO Jurkat cells did not lead to any increase in DNA repair capacity (Fig. 6h), further supporting the notion that GFI1 is required to mediate the activity of PRMT1.

Altogether our results strongly support a mechanism whereby GFI1 promotes DNA repair by mediating PRMT1-dependent ADMA of a specific set of DNA repair proteins such as MRE11 and 53BP1.

## Discussion

We report here that T lymphocytes require the nuclear zinc finger protein GFI1 to efficiently repair DSBs by HR, as supported in multiple systems by Comet and plasmid-based assays as well as γH2AX immunofluorescence. We propose that GFI1 is required for proper interaction between PRMT1 and both MRE11 and 53BP1, and efficient arginine dimethylation of these latter two proteins on their GAR motifs, prior to the occurrence of DNA damage, thus priming them for an optimally rapid response to genotoxic stress. Notably, GFI1 does not affect overall ADMA levels, supporting the notion that its effect is specific to a limited subset of PRMT1 targets.

We note that our results showing defective HR-directed repair and radiosensitivity in Gfi1-deficient cells are similar to those obtained in previous studies using cells expressing a non-methylatable MRE11 with arginine to lysine (R/K) replacements in its GAR motif[19]. Indeed, our own work with this model system showed defects in DNA repair similar to that of GFI1 KO cells.

It is worth mentioning that while we identify an interaction between GFI1 and the ATM kinase, our results suggest that the role of GFI1 in DNA repair is not mediated through ATM as the activity of ATM is not affected by GFI1 deficiency. It is possible that ATM plays a role upstream of GFI1 in DNA repair, or that they interact in some other context unrelated to methylation by PRMT1.

Importantly, the use of an inhibitor of PRMT1 and of a PRMT1 siRNA in cells expressing normal or elevated levels of GFI1 showed that its activity is mediated through PRMT1. The fact that re-expression of GFI1 mutant constructs could rescue the DNA repair phenotype of GFI1 KO cells based on their ability to interact with PRMT1 supports this as well.

Furthermore, the fact that treatment of GFI1 KO cells with the PRMT1 inhibitor or with PRMT1 siRNA did not further impair the survival of these cells following IR exposure or their ability to repair DNA damage clearly shows that PRMT1's activity in the DNA damage response is significantly compromised in GFI1 KO cells and implies that GFI1 is a critical regulator of PRMT1's activity in DNA repair. This is further supported by the observation that PRMT1 overexpression improved DNA repair efficiency in GFI1-expressing cells but not in GFI1 KO cells. Interestingly, our results suggest a mechanism of regulation of PRMT1 where adaptor proteins, such as GFI1, can regulate different activities of the methyltransferase by mediating the interaction of PRMT1 with specific subsets of its targets. Further experiments will show whether this principle is also true for other GFI1-expressing hematopoietic cells, but also sensory epithelial cells, cells of the nervous system and others that are positive for GFI1.

Interestingly, while GFI1 is mainly expressed in lymphoid and myeloid cells, DSB repair is common to all cell types. This suggests that the observed role of GFI1 in DNA repair is fulfilled through different mechanisms in those cell types that do not express GFI1. It remains to be shown which other proteins would fill this role in those cells that do not express GFI1.

Notably, while GFI1 deficiency also leads to a reduction in ADMA on 53BP1, which promotes NHEJ, we did not observe abrogation of this repair pathway in GFI1 KO cells. On the other hand it is unlikely that GFI1 is involved only in HR repair, given that comet assays performed in populations where a majority of cells are in G1 show a substantial delay in repair. Furthermore, analysis of GFI1's effect on γ-H2AX signaling clearly showed GFI1 having an effect in cells in G1 phase, where HR repair is not used. It is possible that the NHEJ plasmid-based assay used here was not sensitive enough to detect the effect of GFI1 deficiency on this pathway, or that other repair pathways also depend on GFI1. In addition, it has been reported in the literature that 53BP1 plays a specific role in DSB repair in heterochromatin, including in HR repair of DSBs in G2, through the regulation of KAP-1 accumulation at break sites[39,40], which represents an additional way in which the effect of GFI1 on 53BP1 methylation may affect DNA repair.

Although we demonstrate a direct interaction between GFI1 and the DDR machinery and that GFI1-mediated post-translational modification of MRE11 is critical in this respect, we cannot exclude the possibility that GFI1 plays a transcriptional role in the cellular response to DNA damage by regulating the expression of DDR genes.

The function of GFI1 described here is reminiscent of another previously reported non-transcriptional role for this protein, i.e., in the regulation of p53 methylation via the recruitment of the demethylase LSD1 to p53. The fact that GFI1-deficient cells exhibit increased p53 activity[4], along with reduced DNA repair capacity as described here, may provide an additional explanation as to why such cells are more sensitive to IR and genotoxic

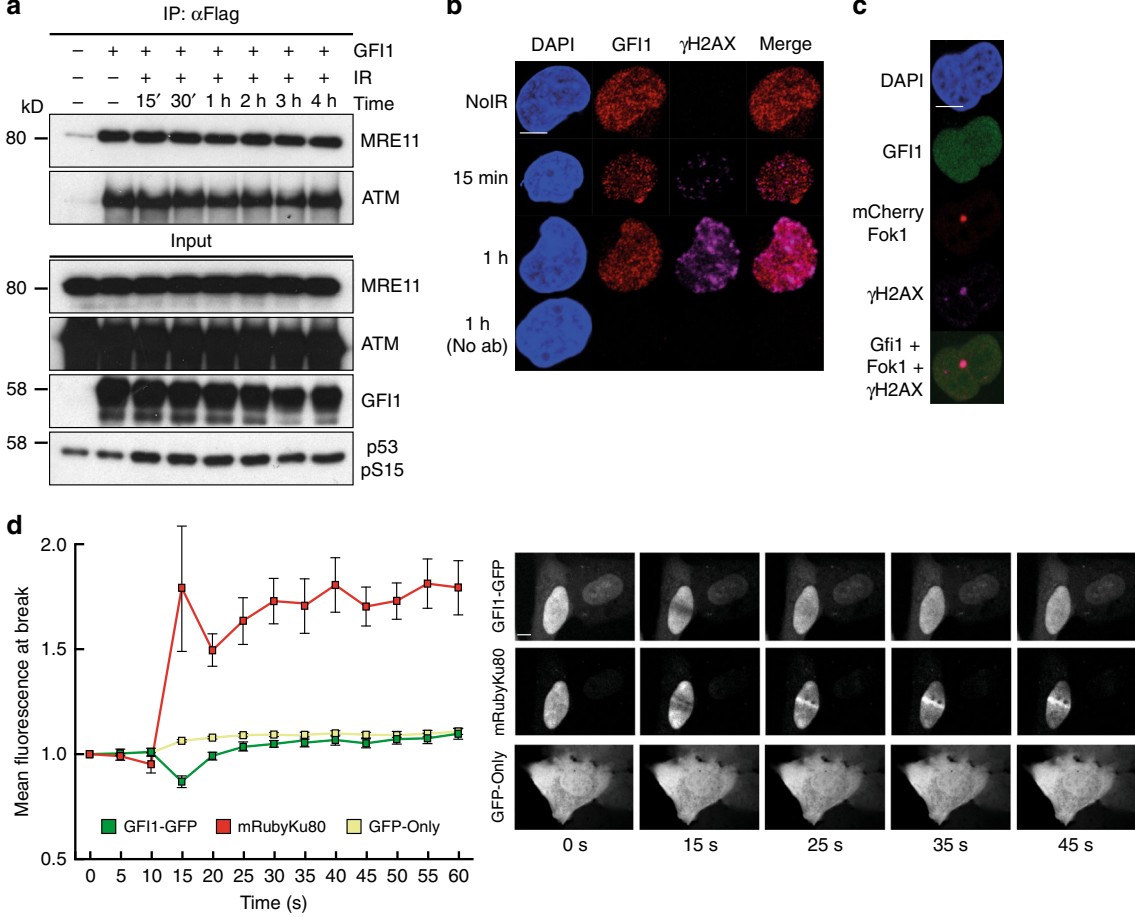

**Fig. 4** GFI1 activities are independent of DNA damage. **a** GFI1-Flag fusion protein was immunoprecipitated in 293T cells treated with 5 Gy IR and allowed to recover for the indicated amount of time. Extracts were separated by SDS–PAGE and blotted for the indicated proteins. **b** SupT1 cells were spread on glass slides 15 min and 1 h after irradiation using a Cytospin, stained for endogenous Gfi1 and γ-H2AX and visualized for immunofluorescence by confocal microscopy. Control cells stained without primary antibody but with secondary antibodies are shown. **c** U2OS cells carrying a LacO array and expressing a LacR-Fok1-mCherry endonuclease were transfected with a vector expressing the GFI1-GFP fusion protein. These cells were plated on cover glass, stained for γ-H2AX and visualized for immunofluorescence by confocal microscopy. **d** U2OS cells expressing a GFI1-GFP fusion protein were exposed to 405 nm UV micro-irradiation and the recruitment of the GFI1-GFP fusion protein to the site of damage was quantified by confocal microscopy. Average signal intensity is shown with error bars representing s.d. Recruitment of Ku80-mRuby2 fusion protein and GFP protein are shown as controls. Representative images of selected time points are shown on the right. Scale bar represents 10 µm

chemotherapeutics such as cytarabine. Conversely, the elevated expression of GFI1 could explain chemo- or radio-resistance of lymphoid leukemia or lymphoma, as we observed here and have implications for treatment response in other tumor types that overexpress GFI1, including medulloblastomas and neuroendocrine lung carcinomas. It also stands to reason that tumors which display reduced levels of GFI1, as is the case for some acute myeloid leukemias[32], may be subject to increased genomic instability and thus greater sensitivity to radiation therapy and chemotherapy.

## Methods

**Mouse strains**. *Gfi1* KO, *GFI1* KI, *Gfi1* KD mice used in this study, have been previously described[31,41]. Mice have been bred on to C57BL/6 genetic background and were maintained in a specific-pathogen-free plus environment at the Institut de Recherches Cliniques de Montreal (IRCM). The Institutional Review Board of the IRCM approved all animal protocols and experimental procedures were performed in compliance with IRCM and CCAC (Canadian Council of Animal Care) guidelines.

**Cell culture**. SupT1 (ATCC CRL-1942) and Jurkat (ATCC TIB-152) cells were maintained in RPMI media (Multicell) supplemented with 10% Bovine Growth Serum (RMBIO Fetalgro) and 100 IU Penicillin and 100 µg/ml Streptomycin

(Multicell). We verified that none of the cell lines used in this study were found in the Register of Misidentified Cell Lines maintained by the International Cell Line Authentication Committee (http://iclac.org/databases/cross-contaminations/).

All cell lines used were tested and shown to be negative for mycoplasma contamination using both immunofluorescence with DAPI staining and PCR amplification using the following primer mix (Christian Praetorius, https://bitesizebio.com/23682/homemade-pcr-test-for-mycoplasma-contamination/, 2015)[42,43]:

Forward Primers: Myco-5-1 CGCCTGAGTAGTACGTTCGC, Myco-5-2 CGCCTGAGTAGTACGTACGC, Myco-5-3 TGCCTGAGTAGTACATTCGC, Myco-5-4 TGCCTGGGTAGTACATTCGC, Myco-5-5 CGCCTGGGTAGTACATTCGC, Myco-5-6 CGCCTGAGTAGTATGCTCGC

Reverse Primers: Myco-3-1 GCGGTGTGTACAAGACCCGA, Myco-3-2 GCGGTGTGTACAAAACCCGA, Myco-3-3 GCGGTGTGTACAAACCCCGA

**Viable cell counts**. SupT1 or Jurkat cells were seeded in triplicate at 1 million cells per ml in 1 ml of RPMI media in 24 well plates. Cells were treated with IR or Cytarabine on day 0 and each replicate was counted in duplicate using a haemocytometer on days 1 through 3.

**Immunofluorescence**. Cells were centrifuged onto glass slides using a Shandon Cytospin 4 at 400 rpm (~35×g) for 2 min at the lowest acceleration and deceleration settings and fixed for 10 min in 4% PFA. The cell membrane was solubilized in PBS containing 5% FBS and 0.5% Triton X-100. Samples were incubated for 1 h in solubilizing solution containing primary antibodies. Secondary detection was done with Alexa Fluor-488 conjugated, 546 conjugated or 647 conjugated antibodies

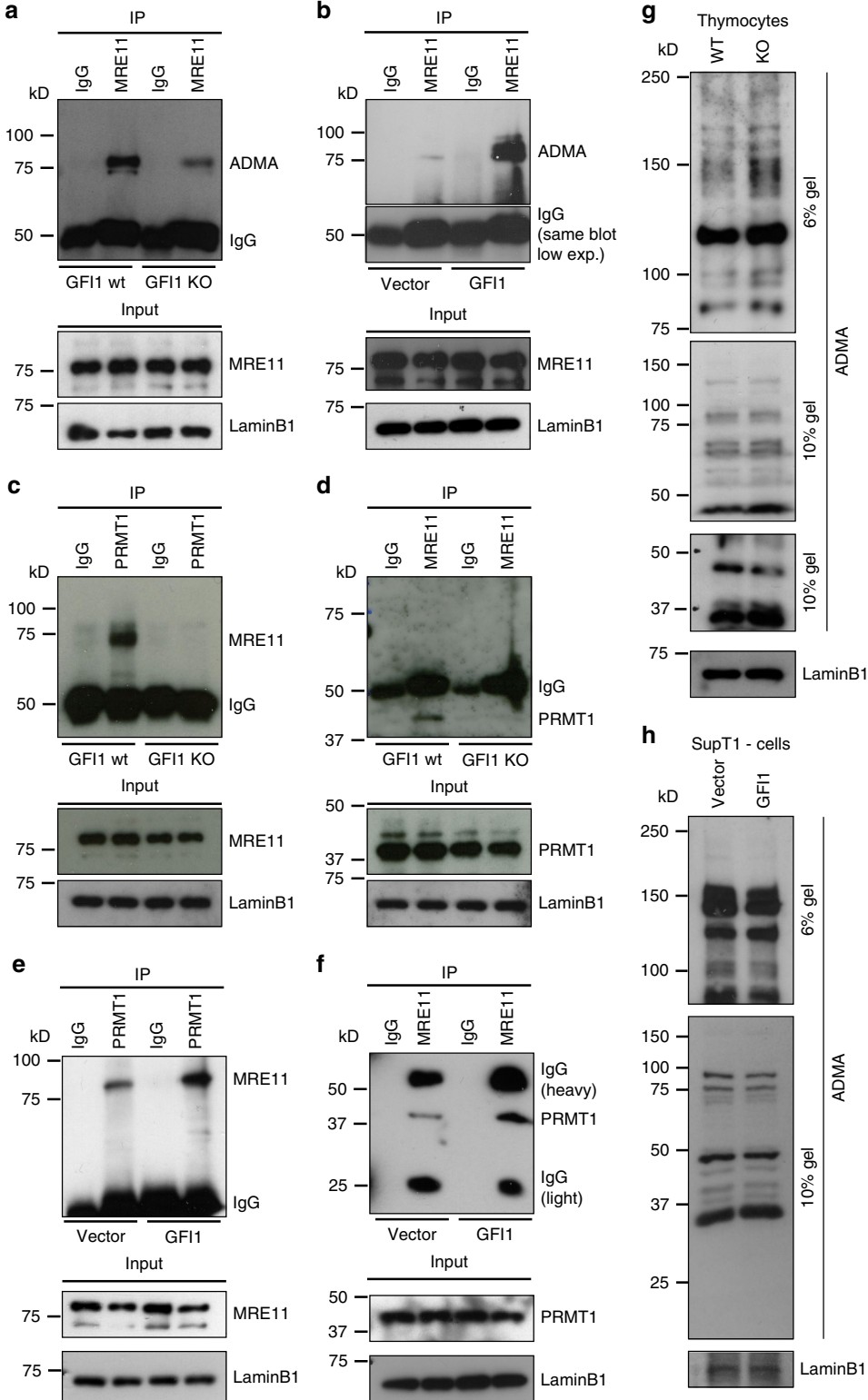

**Fig. 5** GFI1 mediates PRMT1-dependent methylation of MRE11. **a** Nuclear extracts were prepared from thymocytes extracted from *Gfi1* WT mice and matching *Gfi1* KO mice. MRE11 was immunoprecipitated from these extracts, proteins separated by SDS–PAGE and blotted for ADMA. Control blots on input cell extract are shown below. **b** Nuclear extracts were prepared from SupT1 cells overexpressing GFI1 and Vector control cells and immunoprecipitated for MRE11, separated by SDS–PAGE and blotted for ADMA. **c** Extracts prepared as described in **a** were immunoprecipitated for PRMT1, separated by SDS–PAGE and blotted for MRE11. **d** Extracts prepared as described in **a** were immunoprecipitated for MRE11, separated by SDS–PAGE and blotted for PRMT1. **e** Extracts prepared as described in **b** were immunoprecipitated for PRMT1, separated by SDS–PAGE and blotted for MRE11. **f** Extracts prepared as described in **b** were immunoprecipitated for MRE11, separated by SDS–PAGE and blotted for PRMT1. **g** Nuclear extracts from thymocytes extracted from *Gfi1* WT mice and matching *Gfi1* KO mice were separated by SDS–PAGE and blotted for ADMA. **h** Nuclear extracts were prepared from SupT1 cells overexpressing GFI1 and Vector control cells and were separated by SDS–PAGE and blotted for ADMA

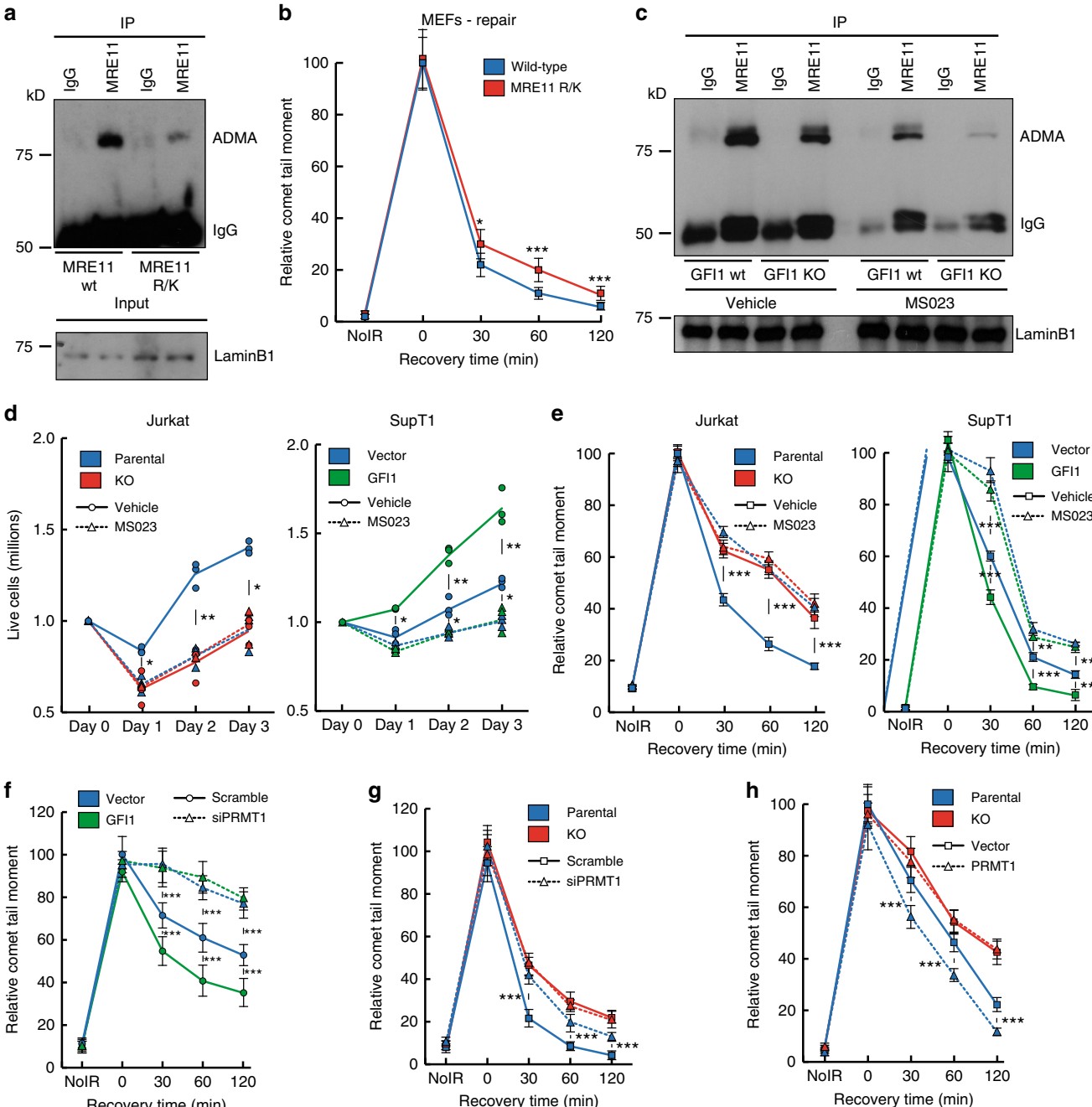

Fig. 6 GFI1's role in DNA Repair is mediated through PRMT1 activity. **a** Nuclear extracts were prepared from MEFs expressing an R/K mutant form of MRE11 and wild-type control cells. Extracts were immunoprecipitated for MRE11 and blotted for ADMA. **b** MEFs expressing an R/K mutant form of MRE11 and wild-type control cells were exposed to 5 Gy IR and allowed to recover for the indicated time. Cells were then lysed and analyzed by alkaline Comet assay. Comet tail moment averages are shown. One of three replicate experiments is shown. Error bars represent s.d. *$p < 0.05$, **$p < 0.01$, ***$p < 0.001$ on a Welch corrected *T*-test. **c** GFI1 KO Jurkat cells and parental control cells were treated for 48 h with 500 nM MS023 inhibitor. Nuclear extracts were then immunoprecipitated for MRE11 and blotted for ADMA. **d** GFI1 KO Jurkat cells and parental control cells (left) and SupT1 overexpressing GFI1 and vector control cells (right) were pre-treated with 500 nM of MS023 inhibitor or vehicle were seeded at 1 million cells per ml and exposed to 5 Gy IR. Cells were counted each following day. Dashed lines show average cell numbers and individual data points of a triplicate experiment are shown. **e** Cells as in **d** were exposed to 5 Gy IR and allowed to recover for the indicated time. Cells were then lysed and analyzed by Comet assay as in **b**. **f** SupT1 overexpressing GFI1 and vector control cells were electroporated with an siRNA against PRMT1. Electroporated cells were FACS sorted 24 h later. 24 additional hours later, cells were exposed to 5 Gy IR, allowed to recover for the indicated time, then lysed and analyzed by alkaline Comet assay as in **b**. **g** GFI1 KO Jurkat cells and parental control cells were treated as in **f** and analyzed by alkaline Comet assay as in **b**. **h** GFI1 KO Jurkat cells and parental control cells were electroporated with a pcDNA3.1 plasmid expressing PRMT1 or a vector control. Electroporated cells were FACS sorted 24 h later. 24 additional hours later, cells were exposed to 5 Gy IR, allowed to recover for the indicated time, then lysed and analyzed by alkaline Comet assay as in **b**

(Thermo Fisher) and cells were counterstained with 4',6-diamidino-2-pheny-lindole. Microscopy was carried out on a Zeiss LSM 700 microscope.

**Flow cytometry**. Cells were fixed for 10 min in 4% PFA and solubilized in PBS containing 5% FBS and 0.5% Triton X-100. γ-H2AX staining in thymocytes was done using either Alexa-488 or Alexa-647 conjugated antibody and in SupT1 using an unconjugated primary antibody and a 647 conjugated secondary antibody. Propidium Iodide Staining was used as a cell cylce marker where indicated. Fluorescence signal was measured on a FACSCalibur (BD Biosciences).

**Cell cycle analysis**. Cells were fixed in 75% EtOH and stored at − 20 °C overnight. The cells were centrifuged washed in PBS and resuspended in 200 µl of PBS containing 100 µg RNAse and 1 µg Propidium Iodine. Samples were incubated for 15 min and analyzed using a FACSCalibur. Cell Cycle profiles were analyzed using FlowJo (Tree Star Software).

**Single cell electrophoresis (comet assay)**. Single strand electrophoresis (comet assays) experiments were based on the procedure described by Olive et al[44]. Microscopy slides (Ultident cat. No 170-7107A-S) were coated with 800 µl of 1% low melt agarose (Sigma, A9045) 24 h prior to the assay. 200 µl of cells at a concentration of 50,000 cells per ml were mixed to 600 µl of molten 1% low melt agarose, allowed to solidify for 2 min before being submerged in lysis buffer (Alakaline: 1.2 M NaCl, 100 mM Na₂EDTA, 0.1% sarkosyl, 0.26 M NaOH, pH > 13; Neutral: 2% sarkosyl, 0.5 M Na₂EDTA, 0.5 mg/ml proteinase K, pH 8) for 15–18 h at 4 °C for alkaline lysis and for 4 h at 37 °C for neutral lysis. Slides were washed twice in rinsing buffer (Alkaline: 0.03 M NaOH, 2 mM Na₂EDTA, pH ~12.3; Neutral: 90 mM Tris, 90 mM Boric acid, 2 mM Na₂EDTA, pH 8.5) before electrophoresis (60 mA, for 12 min, in rinsing buffer). Slides were then rinsed in water and stained with Propidium Iodide (Staining) and rinsed in water. Microscopy was carried out on a Zeiss LSM 700 microscope. Comet tail moments were measured using the CometScore software (TriTeck Corp).

**Co-immunoprecipitation**. For each Immunoprecipitation, 10 million cells were lysed in buffer I (0.5% NP-40, 10 mM Hepes, 10 mM KCl, 2 mM EDTA, 10% Glycerol, Complete protease inhibitor (Roche), pH 7.5), incubated on ice for 10 min and centrifuged for 10 min at 18,000×g. Pellets were lysed in 500 µl buffer II (50 mM Sodium Phosphate, 300 mM NaCl, 1 mM β-mercaptoethanol, 10% Glycerol, 0.5% NP-40, 0.5% Triton X-100, Complete protease inhibitor (Roche), pH 7.5), mixed by vortexing and sonicated twice on a Brason digital sonifer for 10 s at 50% output followed by 10 min incubation on ice and centrifuged for 10 min at 18,000×g. Supernatant was incubated for 2 h using the antibody of interest followed by 1 h incubation with protein-A or protein-G agarose beads (Roche). Beads were washed 4 times with buffer II and proteins were extracted by boiling the beads for 5 min in SDS–PAGE sample loading buffer prior to separation by SDS–PAGE and transfer to PVDF membranes for blotting (see Antibodies section below). Uncropped images of all blots are shown in Supplementary Figures 13–21.

**Antibodies**. The following antibodies were used:

For immunoprecipitation: MRE11 ab109623, 1 µl (Abcam); PRMT1 ab73246, 1 µl (Abcam); 53BP1 A300-272A, 1 µl (Bethyl); ATM ab32420, 3 µl (Abcam); GFI1 AF3540 2 µl (R&D Systems).

For immunoblotting: MRE11 ab109623, 1:10,000 (Abcam); PRMT1 ab12189, 1:2,000 (Abcam); 53BP1 A300-273A, 1:5000 (Bethyl); ATM ab32420, 1:5000 (Abcam); LaminB1 b-10 (mouse target only), 1:5000 (Santa-Cruz) or LaminB1 ab16048 (mouse and human target), 1:10,000 (Abcam); Asymmetric di-methyl arginine (ADMA) Asym-26 1:2000 (Epicypher); GFI1 AF3540 1:2000 (R&D Systems).

For immunofluorescence and FACS: γ-H2AX S139, 1:200 (Cell Signalling); ATM pS1981, 1:500 (Rockland 200-301-400); 53BP1 A300-272A, 1:500 (Bethyl); MRE11 nb100-142 1:500 (Novus Biologicals).

**Mass spectrometry analysis**. The in-gel digestion protocol is based on the results obtained by Havlis et al[45]. Gel bands were excised under a clean bench and each band was cut in 1 mm³ pieces. For the following steps, all volumes were adjusted according to the volume of gel pieces. Gel pieces were first washed with water for 5 min and destained twice with the destaining buffer (100 mM sodium thiosulfate, 30 mM potassium ferricyanide) for 15 min. An extra wash of 5 min was performed after destaining with a buffer of ammonium bicarbonate (50 mM). Gel pieces were then dehydrated with acetonitrile. Proteins were reduced by adding the reduction buffer (10 mM DTT, 100 mM ammonium bicarbonate) for 30 min at 40 °C, and then alkylated by adding the alkylation buffer (55 mM iodoacetamide, 100 mM ammonium bicarbonate) for 20 min at 40 °C. Gel pieces were dehydrated and washed at 40 °C by adding ACN for 5 min before discarding all the reagents. Gel pieces were dried for 5 min at 40 °C and then re-hydrated at 4 °C for 40 min with the trypsin solution (6 ng/µL of trypsin sequencing grade from Promega, 25 mM ammonium bicarbonate). The concentration of trypsin was kept low to reduce signal suppression effects and background originating from autolysis products when performing LC-MS/MS analysis. Protein digestion was performed at 58 °C

for 1 h and stopped with 15 µL of 1% formic acid/2% acetonitrile. Supernatant was transferred into a 96-well plate and peptides extraction was performed with two 30-min extraction steps at room temperature using the extraction buffer (1% formic acid/50% ACN). All peptide extracts were pooled into the 96-well plate and then completely dried in vacuum centrifuge. The plate was sealed and stored at −20 °C until LC-MS/MS analysis.

Prior to LC-MS/MS, protein digests were re-solubilized under agitation for 15 min in 10 µL of 2%ACN/1% formic acid and some bands were pooled together. The LC column was a C18 reversed phase column packed with a high-pressure packing cell. A 75 µm i.d. Self-Pack PicoFrit fused silica capillary column (New Objective, Woburn, MA) of 15 cm long was packed with the C18 Jupiter 5 µm 300 Å reverse-phase material (Phenomenex, Torrance, CA). This column was installed on the Easy-nLC II system (Proxeon Biosystems, Odense, Denmark) and coupled to the LTQ Orbitrap Velos (ThermoFisher Scientific, Bremen, Germany) equipped with a Proxeon nanoelectrospray ion source. The buffers used for chromatography were 0.2% formic acid (buffer A) and 100% acetonitrile/0.2% formic acid (buffer B). Two different peptide separation gradients were used according to the molecular weight of the proteins. For the high molecular weight proteins, 5 µL of sample were loaded on column at a flow rate of 600 nL/min and, subsequently, the gradient went from 2–40% buffer B in 43 min and then from 40–80% buffer B in 16 min at a flow rate of 250 nL/min. For the low molecular weight proteins, the gradient went from 2–40% buffer B in 10 min and then from 40–80% buffer B in 4 min at a flow rate of 600 nL/min. LC-MS/MS data acquisition was accomplished using a seven scan event cycle comprised of a full scan MS for scan event 1 acquired in the Orbitrap. The mass resolution for MS was set to 60,000 (at m/z 400) and used to trigger the six additional MS/MS events acquired in parallel in the linear ion trap for the top ten most intense ions. Mass over charge ratio range was from 360 to 2000 for MS scanning with a target value of 1,000,000 charges and from ~1/3 of parent m/z ratio to 2000 for MS/MS scanning with a target value of 10,000 charges. The data dependent scan events used a maximum ion fill time of 100 ms and 1 microscan. Target ions already selected for MS/MS were dynamically excluded for 15 s. Nanospray and S-lens voltages were set to 1.5 kV and 50 V, respectively. Capillary temperature was set to 225 °C. MS/MS conditions were: normalized collision energy, 35 V; activation q, 0.25; activation time, 10 ms.

Proteomics data were analyzed with Crapome[46], an online analytical resource for the identification of nonspecific interactions from multiple Affinity Purification Mass Spectrometry (AP-MS) studies. Once loaded into Crapome (Human version 1.1.), our proteomics data were compared against selective controls (CC62, CC63 and CC66) that offers the same conditions as our AP-MS analysis which involved: HEK293 cells, agarose, M2 anti-FLAG and LTQ Orbitrap. SAINT score and BFDR were then calculated with SAINTexpress[47] by involving the controls enumerated above. Prey abundance were also normalized by applying the NSAF[48] (Normalized Spectral Abundance Factor) method. NSAF is calculated as the number of peptides per protein (Sp), divided by the protein's length (L), divided by the sum of Sp/L of all proteins in a given experiment.

**Laser track irradiation**. For 405-nm UV laser irradiation, experiments were performed as described by Klement et al[37]. U2OS cells pre-treated with 2 µM Hoechst 33342 (Sigma-Aldrich) for 5 min were imaged at 37 °C using a custom-built microscope (Cell Observer; Carl Zeiss/Intelligent Imaging Innovations), equipped with a heated CO2 incubator, diode-based lasers (405, 488, 561, and 633 nm), and a spinning-disk confocal scanning unit (CSU-X1; Yokogawa Electric Corporation) using a ×40, 1.4 NA immersion oil objective lens. UV laser damage was induced by a 100-mW, 405-nM diode laser using a Vector Scan Unit (Intelligent Imaging Innovations) where the effective light output was measured as ~8 mW at the objective when using 100% power. A single line scan of the 405-nm laser at 70% power was sufficient to generate DNA DSBs as demonstrated by the rapid recruitment of KU70[36], which was estimated to be equivalent to ~40–60 Gy cellular dose.

**Homologous recombination repair assay**. Cells were electroporated using a Neon Transfection System (Thermo Fisher Scientific) with 3 pulses of 1350 v, 10 ms in a 100 µl tip. Cells were electroporated with the pNLS-iRFP670 plasmid (Addgene #45466) as a positive control for electroporation, pCR2.1 Clover Lamin Donor as a repair template and one of pX330-LMNA1 or pX330-LMNA2 plasmids expressing Cas9 and a gRNA for the Lamin A locus[33]. Cells were analyzed by FACS 72 h following electroporation. Live cells were gated using the forward scatter vs. side scatter plot and electroporated cells were gated using the iRFP670 signal. Cells positive for HR repair were gated using the Clover signal with control cells electroporated without a gRNA as negative controls.

**Non-homologous end joining repair assay**. Cells were electroporated as for the HR assay with the pNLS-iRFP670 plasmid, EJ5-GFP plasmid and a plasmid directing the expression of the I-SceI endonuclease[34]. Cells were analyzed by FACS 72 h following electroporation. Live cells were gated using the forward vs. side scatter and electroporated cells were gated using the iRFP670 signal. Cells positive for NHEJ repair were gated using the GFP signal with control electroporated without the I-SceI expression vector as negative controls.

**Scoring technique for chromosome breakage analysis**. Analysis was performed on 50 Giemsa-stained metaphases for each tested specimen with three or more specimen per genotype. The number and type of structural chromosome anomalies were scored. Chromatid gaps were not included in the final score. Isochromatid gaps, chromatid and isochromatid breaks, deletions and fragments were scored as a single break. Structural rearrangements including dicentrics, rings and radial figures were scored as two breakage events. The mean number of breaks per cell was scored for each sample.

**In vitro methylation**. GST-MRE11-GAR or GST control proteins were incubated with purified PRMT1 protein for 1 h at 37 °C in the presence of between 0 to 10 µg of GFI1 protein and S-[Methyl-3H]. Reactions were separated on a 12% SDS–PAGE gel, dried and exposed on film overnight at −80 °C.

**PRMT1 Inhibition with MS023**. The MS023 inhibitor was purchased from Cayman Chemical (#18361) and resuspended in DMSO. The inhibitor was used at a final concentration of 500 nM and DMSO alone was used as a vehicle control.

**GFI1 rescue experiments**. Jurkat GFI1 KO cells were electroporated using a Neon Transfection System (Thermo Fisher Scientific) with 3 pulses of 1350 v, 10 ms in a 100 µl tip. Cells were transfected with 1 µg iRFP plasmid as a transfection control and 2 g of a plasmid expressing GFI1 variant constructs. 24 h after electroporation, cells were sorted for iRFP positive cells. Sorted cells were used in comet assays after an additional 24 h.

**PRMT1 siRNA treatment**. Cells were electroporated using a Neon Transfection System (Thermo Fisher Scientific) with 3 pulses of 1350 v, 10 ms in a 100 µl tip. Cells were transduced with an siRNA against PRMT1 targeting the sequence 5′-CGTCAAAGCCAACAAGTTA-3′ or a non-targeting control (Negative Control NC1, Integrated DNA Technologies Cat. No. 51-01-14-03) and a TYE 563 DS transfection control (Integrated DNA Technologies Cat No. 51-01-14-03).

24 h after electroporation, cells were sorted for 563 positive cells. Sorted cells were used in comet assays after an additional 24 h.

**PRMT1 overexpression**. Jurkat GFI1 KO cells were electroporated using a Neon Transfection System (Thermo Fisher Scientific) with 3 pulses of 1350 v, 10 ms in a 100 µl tip. Cells were transfected with 1 µg iRFP plasmid as a transfection control and 2 µg of a plasmid expressing PRMT1. 24 h after electroporation, cells were sorted for iRFP positive cells. Sorted cells were used in comet assays after an additional 24 h.

**Statistical analysis**. The statistical significance of results for Comet assays, cell growth counts, immunofluorescence foci counts and FACS analysis of γ-H2AX signal was tested using a two-tailed Welch corrected Student's T-test. The significance of Annexin V staining results was tested using a two-tailed Fisher's exact test. The sample size of data points for each assay is shown in Supplementary Data 2. Loading controls for PRMT1, MRE11 and 53BP1 in Co-IP experiments were quantified using ImageJ and the aggregated results are presented in Supplementary Figure 12.

**Data availability**. The raw proteomics data, which are presented in Fig. 3, have been uploaded to the PRIDE archive and are available under accession number: PXD008897. The data that support the findings of this study are available from the corresponding author upon request.

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

## Acknowledgements

We thank: The staff of the IRCM Animal Facility, microscopy platform and Flow Cytometry facilities; Denis Faubert and the IRCM Proteomics discovery platform; Sylvie Lavallée and the staff of the Quebec Leukemia Cell Bank and Ludivine Litzler for discussion and suggestions. This work was supported by a CIHR (Canadian Institutes of Health Research) Foundation grant (FGN-148372) and a Canada Research Chair Tier 1 to T.M. C.V. was funded by a post-doctoral fellowship from FRQS (Fonds de Recherche en Santé du Québec) and by a post-doctoral fellowship from CIHR.

## Author contributions

Conceptualization, C.V. and T.M.; investigation, C.V., R.C., J.F., Z.Y., J.B., J.P. and D.F.; resources, C.K., J.H., G.D., J-F C., S.R., A.O., T.M. writing—review and editing, C.V., J.H., G.D., S.R., A.O., E.D. and T.M.

## Additional information

**Competing interests:** The authors declare no competing interests.

