## [Peer Review File · Nature Communications]

Reviewers' comments:

Reviewer #1 (Remarks to the Author):

In this manuscript, Vadnais and colleagues explore the role of the transcription factor GFI1 in DNA repair. Having previously showed that GFI1 deletion leads to hypersensitivity to gamma-irradiation, they use several engineered cell models, to show that GFI1 is needed for efficient double-strand break repair through the homologous recombination pathway. They identify Mre11 and 53BP1 as being GFI1 interacting partners, as well as PRMT1 and demonstrate that GFI1 depletion impairs the arginine methylation of these DNA repair proteins by PRMT1. They propose that GFI1 plays a novel, non-transcriptional role as a critical accessory protein regulating the interaction between PRMT1 and a subset of its substrates.

The authors use a type I PRMT inhibitor, which impairs the proliferation of GFI1-overexpressing cells, a phenotype they attribute to less efficient DNA repair. However, the inhibitor they use is not specific to PRMT1, as it targets all type I PRMTs. To clearly demonstrate that the effect of GFI1 on DNA repair is PRMT1-dependent they need to specifically target PRMT1 in their systems (e.g. by shRNA or CRISPR/Cas9).

Their data argue that even in cells with 'normal' levels of GFI1, type I PRMT inhibition blocks cell proliferation (see Fig. 6d) and delays DNA damage recovery time (Fig 6.e); this argues against a critical role of GFI1 expression levels in determining PRMT-dependent DNA repair.

Major:

1) Specific targeting of PRMT1 in the context of GFI1 overexpression (ideally using both their SupT1 cell line and the GFI1-KI thymocytes) is needed to ascertain whether this methyltransferase is critical for the GFI1-dependent DNA repair phenotypes they observe. Similarly, would overexpressing PRMT1 in GFI1 knockdown cells (Jurkat 'KO') rescue the DNA repair defects, as an interaction between MRE11 and PRMT1 is still detected in these cells (see Fig. suppl. 5a)?

2) The central hypothesis states that GFI1 is an accessory/scaffold protein for PRMT1 to efficiently methylates its substrates yet 53BP1 was not identified by mass spec as a GFI1 interacting protein, unlike MRE11. The authors should comment on that.

3) In the same vein, although in absence of GFI1 there is a clear abrogation of the MRE1/PRMT1 interaction (Fig. 5c&d), the data presented in Fig suppl. 4 is not convincing at all. Unlike MRE11, 53BP1 binding to PRMT1 seems to be marginally affected by GFI1 knockout. The western-blot are of poor quality and overall not conclusive. Perhaps light chain specific secondary antibodies should be used when looking at PRMT1, as it is very close in size to IgG heavy chain (Fig. 3i, 5d, suppl. 4d, suppl. 5b).

4) There is no description of the mass spec experiment in the methods. In the supplemental table 1, what do the 3 'bands' correspond to? Where is the number of assigned spectra for the control sample (IP with extracts from 293T non-transfected with Flag-GFI1)? Except for the well-known interactors of GFI1, the number of spectra and the protein score are both very low for most of the newly identified proteins, including PRMT1, which might explain the very weak binding observed between GFI1 and PRMT1 (see Fig. 3i). In general, spectral counting is not an appropriate quantification method, as it is poorly accurate and very biased towards larger proteins. A volcano plot that displays the fold change and the p-value of the quantified proteins as compared to control (e.g. IgG or Flag + no GFI1 transfection) is a better way of representing mass spec data, and enables the identification of true targets and not false positive or recurrent contaminants.

5) The IC50 of MS023 needs to be determined with the two cell lines tested. A biological marker showing MS023 activity in cells should also be included in the last figure, e.g. a H4R3me2a or H3R2me2a western-blot with increasing doses of the inhibitor, or an ADMA WB with the full blot. 500uM seems to be quite a high dose since others have shown that the IC50 is in the nanomolar range for MCF7 or 239T cells (Eram et al, ACS Chem Biol, 2016).

Minor:

- 1) Figure 6c: the laminB1 WB needs to be redone
- 2) Error bars in Fig. suppl. 1d & 1e, and Fig. 2c & 2d are missing
- 3) The ATM antibody used in Fig. 3e is not listed in the methods section
- 4) The reference given for the COMET assay ('OB 2006') is not found in the list of references.
- 5) There are a number of typos. For example:
 - a. P.5, "...either at baseline..."
 - b. P.8, "...observed in in the absence..."
 - c. P.10, "...whether they displayed phenotypes consistent phenotypes with those observed..."
 - d. P.11, "...Jurkat cells were also eliminated..."

Reviewer #2 (Remarks to the Author):

Summary

Growth factor independence 1 (GFI1) is a zinc finger transcriptional repressor and master regulator of normal and malignant hematopoiesis. Previous work from Tarik Moroy's group demonstrated that GFI1-deficient leukemia cells are more sensitive to ionizing radiation (IR)-induced apoptosis, mainly because of GFI1's role in transcriptional repression of p53-dependent apoptotic gene expression. This is a follow-up study from the same group aiming to further understand the mechanism by which loss of GFI1 sensitizes cells to IR-induced apoptosis.

In this manuscript, Vadnais et al. report on an additional non-transcriptional role for GFI1 in regulating DNA repair in mouse thymocytes and human leukemia cells by directly modulating the function of DNA damage response proteins, including MRE11 and 53BP1. Using comet assays, γ H2AX immunofluorescence and FACS analysis, the authors found increased and persistent DNA damage in GFI1-deficient cells. By performing HR and NHEJ reporter assays, they determined that GFI1-null cells are defective in HR repair. Mechanistically, the authors found that GFI1 interacts with MRE11 and 53BP1, and with the arginine methyltransferase 1 (PRMT1). It is known that PRMT1 interacts with and methylates MRE11 and 53BP1, an important process for both proteins to function in DNA damage repair. Surprisingly, in GFI1-null cells, the interaction of PRMT1 with MRE11, and with 53BP1 is abolished; the methylation of MRE11 and 53BP1 is reduced, suggesting that GFI1 is essential for maintaining the methylation level of MRE11 and 53BP1. Treatment of cells with the type I methyltransferase inhibitor MS023 eliminated the phenotypic differences between GFI1 WT and KO cells or GFI1-expressing and vector only cells, suggesting that GFI1's role in DNA repair is mediated through PRMT activity.

Overall, the authors convincingly demonstrated the importance of GFI1 in cellular response to DNA damage, which is strongly supported by results from different cell lines and using multiple complementary approaches. Additionally, finding GFI1 as a regulator for PRMT1-mediated methylation of MRE11 and 53BP1 is potentially novel, as it will shed light on the regulation of PRMTs. However, several conclusions are not fully supported by the results. Lack of mechanistic insights in several areas lessened the enthusiasm for this manuscript.

Major concerns:

1. GFI1 interacts with ATM, MRE11 and PRMT1 all through its intermediate domain (Fig. 3e). The authors focus on MRE11 and PRMT1. What about ATM, which interacts with GFI1 even stronger than MRE11 and PRMT1? Could the DNA repair defect in GFI1-null cells be caused by impaired ATM recruitment to DSB sites, or altered kinase activity? Do GFI1, ATM, MRE11 and PRMT1 exist in the same protein complex? Can IP one of them coIP the other three? How does their (in addition to

GFI1 and MRE11) interaction respond to DNA damage (IR)? Does DNA damage induce dissociation of ATM?

2. To determine which DNA repair pathway GFI1 is involved, the authors performed two reporter assays and determined GFI1-deficiency mainly affects HR. As shown in Supplementary Figure 2, more than 90% of thymocytes are in G1 phase, which depends upon NHEJ for repair. In addition to reporter assays, the author should perform FACS analysis or immunofluorescence analysis using cell cycle specific antibody to examine H2AX phosphorylation at different cell cycle stages to further confirm the results from reporter assay.

3. The authors showed that in GFI1-null cells, the interaction of MRE11 and 53BP1 with PRMT1 is abolished and their methylation is reduced. Arginine methylation of MRE11 and 53BP1 is important for their recruitment to DSB sites. The authors should examine if MRE11 and 53BP1 localization to DSB sites is compromised after IR in GFI1-null cells.

4. Loss of GFI1 decreases the methylation of the DDR substrates MRE11 and 53BP1 by abrogating their interaction with PRMT1, suggesting GFI1 may mediate the interaction between PRMT1 and its DDR substrates. Does GFI1 directly interact with PRMT1? Does GFI1 affect PRMT1 activity? In vitro methylation assays of PRMT1 with DDR proteins in the presence or absence of GFI1 should be done. Does GFI1 affect the expression or stability of PRMT1, MRE11, and 53 BP1? Based on the Western blots in Figure 5, it appears that loss of GFI1 decreases the expression of MRE11 and PRMT1.

5. The same group reported that GFI1 promotes leukemia cell survival by restraining p53-mediated transcription activation of apoptotic genes after IR. Does DNA repair defect seen in GFI1-null cells dependent upon p53? The author should knockdown p53 expression and compare DNA repair in wild type and GFI1-null cells. This would help define a non-transcriptional role of GFI1 in DNA repair.

Other concerns:

1. Standard deviation should be used rather than standard error throughout the manuscript.

2. Figure 3a – Proteins should be labeled.

3. Figure 3e- Reciprocal CoIP should be done. In the text it states that only the loss of the intermediate domain (lane 3) affects the interaction between GFI1 and PRMT1 and MRE11 but it appears that the SNAG domain is also important as well as the DNA-binding domain for interaction with MRE11. Blots for ATM, MRE11, and PRMT1 are missing for the input samples.

4. Figure 3i – The endogenous CoIP of GFI1 with PRMT1 is not convincing in both cell lines.

5. Figure 5- in panel b, overexpression of GFI1 enhances MRE11 methylation. This maybe due to more loading of GFI1 overexpressed sample as indicated by more IgG signal. Additionally, does overexpression of GFI1 enhance PRMT1-MRE11 interaction?

6. For comparison of methylation level and the detection of coIP, the authors need to show the amount of target protein being IPed. This is a common issue for Figure 5 (a & b), Figure 6 (a & c), Supplementary Figure 4 (b, c & d), and Supplementary Figure 5 (a, b & c).

7. Figure 6 c – the efficacy of MS023 treatment on cellular ADMA level should be shown. The quality of LaminB1 blot needs to be improved.

Reviewer #3 (Remarks to the Author):

Title: GFI1 is required for efficient DNA Repair by regulating PRMT1 dependent methylation of MRE11 and 53BP1

Summary: The authors present interesting data that implicate GFI1 in the repair of DNA double strand breaks in T cells. Upon the induction of DNA double-strand breaks, GFI1 stimulates the PRMT dependent methylation of MRE11 and 53BP1, thereby promoting repair by homologous recombination. The paper is extremely well written and if the authors can address my concerns, please see the following section, I see no reason why the paper cannot be accepted.

Major concerns:

- 1) The use of the alkaline comet assay for showing effects on DNA double-strand break repair. I am aware that in the text the authors state that this assay is able to detect double- and single-strand breaks, and they are correct. The problem is that the contribution of double-strand breaks to comet length, compared to single-strand breaks, are negligible. The differences in comet length presented in Figures 1, 3 and 6 might be due to differences in single-strand break repair. I would propose the authors repeat one or two of the comet assays, but this time do it under neutral conditions. See whether the differences are still apparent.
- 2) The use of "mean nuclear gamma-H2AX signal" as an indication of double-strand breaks. The "brightness" of a focus is determined by kinase and phosphatase activity, not by the presence of a double-strand break. Irrespective of the brightness, foci number tells you whether there is, or was a double-strand break. Please express the graphs as the number of foci per cell, or an equivalent.
- 3) The specific methylation of MRE11 and 53BP1 are very interesting as these two proteins are the hub for repair pathway choice. Some data are presented that the methylation of MRE11 stimulates homologous recombination, but the methylation of 53BP1 had no effect on non-homologous end joining. Please expand the discussion to include possible reasons why the cell need to methylate both.

Minor points:

- 1) Graphs: I think it is should be a decimal point and not a decimal comma.
- 2) Page 8, line 12. You used "in" twice.
- 3) Page 10, line 17. Remove the word "phenotypes" that appear after the word "consistent".

We would like to thank the reviewers for their insightful comments; below is a point by point response to each comment with our response in blue font and the original comments in black. Each response refers to the corresponding figures and sections of the main text. All sections of the text that have been modified during the review process are highlighted in yellow.

Reviewer #1 (Remarks to the Author):

In this manuscript, Vadnais and colleagues explore the role of the transcription factor GFI1 in DNA repair. Having previously showed that GFI1 deletion leads to hypersensitivity to gamma-irradiation, they use several engineered cell models, to show that GFI1 is needed for efficient double-strand break repair through the homologous recombination pathway. They identify Mre11 and 53BP1 as being GFI1 interacting partners, as well as PRMT1 and demonstrate that GFI1 depletion impairs the arginine methylation of these DNA repair proteins by PRMT1. They propose that GFI1 plays a novel, non-transcriptional role as a critical accessory protein regulating the interaction between PRMT1 and a subset of its substrates.

The authors use a type I PRMT inhibitor, which impairs the proliferation of GFI1-overexpressing cells, a phenotype they attribute to less efficient DNA repair. However, the inhibitor they use is not specific to PRMT1, as it targets all type I PRMTs. To clearly demonstrate that the effect of GFI1 on DNA repair is PRMT1-dependent they need to specifically target PRMT1 in their systems (e.g. by shRNA or CRISPR/Cas9). Their data argue that even in cells with 'normal' levels of GFI1, type I PRMT inhibition blocks cell proliferation (see Fig. 6d) and delays DNA damage recovery time (Fig 6.e); this argues against a critical role of GFI1 expression levels in determining PRMT-dependent DNA repair.

Major:

1) Specific targeting of PRMT1 in the context of GFI1 overexpression (ideally using both their SupT1 cell line and the GFI1-KI thymocytes) is needed to ascertain whether this methyltransferase is critical for the GFI1-dependent DNA repair phenotypes they observe. Similarly, would overexpressing PRMT1 in GFI1 knockdown cells (Jurkat 'KO') rescue the DNA repair defects, as an interaction between MRE11 and PRMT1 is still detected in these cells (see Fig. suppl. 5a)?

In order to confirm that GFI1's role in DNA repair is mediated through PRMT1, we performed siRNA-mediated knockdown of PRMT1 in SupT1 cells expressing either normal or increased levels of GFI1. Following Knockdown of PRMT1, the DNA repair capacity of both cell lines was reduced to an equal level, below that of the non-treated vector cells. These results mirror those obtained using the MS023 inhibitor but are specific to PRMT1 and show that overexpression of GFI1 does not improve DNA repair in the absence of PRMT1, confirming that the role of GFI1 is mediated through it.

To further demonstrate this, we performed siRNA-mediated knockdown of PRMT1 in GFI1 KO Jurkat cells and corresponding WT cells. We found that knockdown of PRMT1 caused a delay in DNA repair in the GFI1 WT cells but caused no further deficiency in the GFI1 KO cells, providing further evidence that GFI1 acts through PRMT1.

Unfortunately, we were unable to perform specific targeting of PRMT1 in our mouse system, as thymocytes cannot be kept in culture long enough to perform an siRNA or shRNA based experiment. Crossing our GFI1 KO mice with existing PRMT1 KO models would have been an alternative solution, but could not be done within the time frame of the reviews.

However, we overexpressed PRMT1 in GFI1 KO and WT Jurkat cells and found that PRMT1 overexpression improved the DNA repair capacity of the GFI1 WT cells but not of the GFI1 KO cells. This suggests that while some interaction between PRMT1 and MRE11 and methylation of MRE11 is present in the GFI1 KO cells, the reduction is sufficient to impair the GFI1-PRMT1 repair axis.

Altogether, these results (presented in Figure 6 f-h, and discussed on page 12, lines 15-26) support the model whereby the DNA repair activity of GFI1 is mediated through PRMT1 specifically, and that conversely, PRMT1 depends on GFI1 for at least part of its role in DNA repair.

2) The central hypothesis states that GFI1 is an accessory/scaffold protein for PRMT1 to efficiently methylates its substrates yet 53BP1 was not identified by mass spec as a GFI1 interacting protein, unlike MRE11. The authors should comment on that.

We believe that the absence of 53BP1 in the Mass Spec experiment may be a false negative. 53BP1 is a large protein, with a predicted weight of 214 kD but frequently migrates at a higher apparent weight and may thus have been excluded from the analysed sample. It is also possible that 53BP1 peptides were not identified for other technical reasons. To support this hypothesis, we have performed Co-IP experiments between GFI1 and 53BP1 and found that in Jurkat and SupT1 cells, immunoprecipitates of endogenous GFI1 can be successfully blotted for 53BP1. These new results (shown in supplementary figure 5d and discussed on page 10, lines 21-24) suggest that GFI1 and 53BP1 do in fact interact.

3) In the same vein, although in absence of GFI1 there is a clear abrogation of the MRE1/PRMT1 interaction (Fig. 5c&d), the data presented in Fig suppl. 4 is not convincing at all. Unlike MRE11, 53BP1 binding to PRMT1 seems to be marginally affected by GFI1 knockout. The western-blotting are of poor quality and overall not conclusive. Perhaps light chain specific secondary antibodies should be used when looking at PRMT1, as it is very close in size to IgG heavy chain (Fig. 3i, 5d, suppl. 4d, suppl. 5b).

We have repeated the IP experiment for the interaction between 53BP1 with a different combination of antibodies (Rabbit for IP, Mouse for blotting). We succeeded to minimize the cross reaction of the blotting antibody with the IgG and can now show much more clearly the decrease in PRMT1 signal following 53BP1 IP in Gfi1 KO cells. The new figure panel is shown in supplementary figure 5c. We also repeated the PRMT1 IP and blotted for 53BP1 but did not obtain significantly better blots.

4) There is no description of the mass spec experiment in the methods. In the supplemental table 1, what do the 3 'bands' correspond to? Where is the number of assigned spectra for the control sample (IP with extracts from 293T non-transfected with Flag-GFI1)? Except for the well-known interactors of GFI1, the number of spectra and the protein score are both very low for most of the newly identified proteins, including PRMT1, which might explain the very weak binding observed between GFI1 and PRMT1 (see Fig. 3i). In general, spectral

counting is not an appropriate quantification method, as it is poorly accurate and very biased towards larger proteins. A volcano plot that displays the fold change and the p-value of the quantified proteins as compared to control (e.g. IgG or Flag + no GFI1 transfection) is a better way of representing mass spec data, and enables the identification of true targets and not false positive or recurrent contaminants.

We have re-analysed the Mass Spec results using more recent methods. The original reference to "bands" in the sample reflected the fact that the gel was cut into separate portions for mass spec analysis. The new analysis method pools the peptides from all sections of gels for analysis. Since we did not originally analyze a control sample with untransfected cells, we re-analyzed the data using Crapome data and used controls performed under similar conditions. This re-analysis eliminated NBS1 and ATM as putative targets, although we had confirmed the interaction with ATM with in-vitro IP experiments. The results are now presented with Fold enrichment values and a Bayesian False Discovery Rate (See Figure 3). We feel that this presentation could replace a Volcano plot depiction.

5) The IC50 of MS023 needs to be determined with the two cell lines tested. A biological marker showing MS023 activity in cells should also be included in the last figure, e.g. a H4R3me2a or H3R2me2a western-blot with increasing doses of the inhibitor, or an ADMA WB with the full blot. 500uM seems to be quite a high dose since others have shown that the IC50 is in the nanomolar range for MCF7 or 239T cells (Eram et al, ACS Chem Biol, 2016).

The concentration used experimentally was in fact 500 nano molar (nM) and not 500 micro molar (μ M). While the figure legend stated this correctly, the materials and methods section was written in a confusing manner and we have modified this to be unambiguous. In addition, we have treated the SupT1 and Jurkat cell lines with increasing concentrations of the MS023 and blotted nuclear extracts for ADMA. We show that there is a significant drop-off of the ADMA signal starting around a 250-500nM concentration. These results are shown in supplementary figure 10, panels a, b, d and e.

Minor:

1) Figure 6c: the laminB1 WB needs to be redone. This LaminB1 blot has been repeated and the figure changed accordingly.

2) Error bars in Fig. suppl. 1d & 1e, and Fig. 2c & 2d are missing.

Supplementary Figures 1d and 1e report the proportion of Annexin V positive cells for the different genotypes from one experiment. The p value is derived from a Fisher's exact test and as such an error bar cannot be presented. However in both cases we show 1 of 3 representative experiments giving similar and significant results.

As for Fig. 2c and 2d, error bars representing standard deviation have been added. These were originally not included because, as these assays are based on FACS, they have large standard deviations but are nonetheless highly significant due to the large sample size used and we initially wished to avoid giving the false impression that the results are not significant. We feel that error bars representing the standard error of the mean would not be a good solution to this issue since the bars are so small as to be invisible, given the large

sample size, and thus uninformative.

3) The ATM antibody used in Fig. 3e is not listed in the methods section.

The antibody used has been added to the Materials and Methods section.

4) The reference given for the COMET assay ('OB 2006') is not found in the list of references. This has been corrected.

5) There are a number of typos. For example:

a. P.5, "...either at baseline..."

b. P.8, "...observed in in the absence..."

c. P.10, "...whether they displayed phenotypes consistent phenotypes with those observed..."

d. P.11, "...Jurkat cells were also eliminated..."

These Typos have been corrected.

Reviewer #2 (Remarks to the Author):

Summary

Growth factor independence 1 (GFI1) is a zinc finger transcriptional repressor and master regulator of normal and malignant hematopoiesis. Previous work from Tarik Moroy's group demonstrated that GFI1-deficient leukemia cells are more sensitive to ionizing radiation (IR)-induced apoptosis, mainly because of GFI1's role in transcriptional repression of p53-dependent apoptotic gene expression. This is a follow-up study from the same group aiming to further understand the mechanism by which loss of GFI1 sensitizes cells to IR-induced apoptosis.

In this manuscript, Vadnais et al. report on an additional non-transcriptional role for GFI1 in regulating DNA repair in mouse thymocytes and human leukemia cells by directly modulating the function of DNA damage response proteins, including MRE11 and 53BP1. Using comet assays, γ H2AX immunofluorescence and FACS analysis, the authors found increased and persistent DNA damage in GFI1-deficient cells. By performing HR and NHEJ reporter assays, they determined that GFI1-null cells are defective in HR repair. Mechanistically, the authors found that GFI1 interacts with MRE11 and 53BP1, and with the arginine methyltransferase 1 (PRMT1). It is known that PRMT1 interacts with and methylates MRE11 and 53BP1, an important process for both proteins to function in DNA damage repair. Surprisingly, in GFI1-null cells, the interaction of PRMT1 with MRE11, and with 53BP1 is abolished; the methylation of MRE11 and 53BP1 is reduced, suggesting that GFI1 is essential for maintaining the methylation level of MRE11 and 53BP1. Treatment of cells with the type I methyltransferase inhibitor MS023 eliminated the phenotypic differences between GFI1 WT and KO cells or GFI1-expressing and vector only cells, suggesting that GFI1's role in DNA repair is mediated through PRMT activity.

Overall, the authors convincingly demonstrated the importance of GFI1 in cellular response to DNA damage, which is strongly supported by results from different cell lines and using multiple complementary approaches. Additionally, finding GFI1 as a regulator for PRMT1-mediated methylation of MRE11 and 53BP1 is potentially novel, as it will shed light on the regulation of PRMTs. However, several conclusions are not fully supported by the results. Lack of mechanistic insights in several areas lessened the enthusiasm for this manuscript.

Major concerns:

1. GFI1 interacts with ATM, MRE11 and PRMT1 all through its intermediate domain (Fig. 3e). The authors focus on MRE11 and PRMT1. What about ATM, which interacts with GFI1 even stronger than MRE11 and PRMT1? Could the DNA repair defect in GFI1-null cells be caused by impaired ATM recruitment to DSB sites, or altered kinase activity? Do GFI1, ATM, MRE11 and PRMT1 exist in the same protein complex? Can IP one of them coIP the other three? How does their (in addition to GFI1 and MRE11) interaction respond to DNA damage (IR)? Does DNA damage induce dissociation of ATM?

A re-analysis of the Mass Spectrometry results showed that ATM is no longer identified as a putative binding partner of GFI1 in that experiment. However, in-vitro experiments conclusively show the interaction between these two proteins

In order to assess whether the DNA repair defects in GFI1 KO cells were related to ATM activity, we performed immunofluorescence assays for the active p-ATM protein and found that GFI1 deficiency had no effect on the appearance or disappearance of p-ATM foci following IR exposure (See supplementary Figure 3 and text on page 9, lines 3-9).

Also, co-immunoprecipitation assays, which had originally been performed but had not been presented in the paper, show that the interaction between GFI1 and ATM is unaffected by exposure to IR, as is the interaction between GFI1 and MRE11 (See Fig. 4 a and text on page 9, lines 17-21).

To test whether GFI1, ATM, MRE11 and PRMT1 exist in the same complex, we have performed co-IP experiments of ATM and found that it interacts with MRE11, but not with PRMT1 (See supplementary Figure 9 c and d, and text on page 11, line 13-17; three independent ATM co-IPs failed to show interaction with PRMT1, one is reported here).

Since our experiments show that GFI1 interacts with all 3 other proteins, but that ATM does not interact with PRMT1, we speculate that GFI1, PRMT1 and MRE11 exist in one complex while other GFI1 proteins and ATM exist in another complex. The interaction between ATM and MRE11 is likely given the known interaction between ATM and the MRN complex.

Altogether, we feel that ATM is unlikely to mediate GFI1's effect on DNA repair efficiency. It is possible however, that ATM plays a role upstream of GFI1 in this whole pathway, or that they interact in the context of a different part of the repair process unrelated to methylation by PRMT1. These implications are mentioned in the discussion (pages 13, lines 16-20).

2. To determine which DNA repair pathway GFI1 is involved, the authors performed two reporter assays and determined GFI1-deficiency mainly affects HR. As shown in Supplementary Figure 2, more than 90% of thymocytes are in G1 phase, which depends upon NHEJ for repair. In addition to reporter assays, the author should perform FACS analysis or immunofluorescence analysis using cell cycle specific antibody to examine H2AX phosphorylation at different cell cycle stages to further confirm the results from reporter assay.

We have performed additional FACS-based γ -H2AX experiments with a Propidium Iodide staining to identify cells in different cell cycle phases. We found that, in thymocytes, the difference in γ -H2AX staining we had previously observed was present in each phase of the cell cycle (See supplementary Figure 2 d and text page 7, lines 6-8). This strongly suggests that, in addition to playing a role in HR repair, GFI1 may play a role in other repair pathways, possibly including NHEJ (See discussion page 14, lines 12-23).

3. The authors showed that in GFI1-null cells, the interaction of MRE11 and 53BP1 with PRMT1 is abolished and their methylation is reduced. Arginine methylation of MRE11 and 53BP1 is important for their recruitment to DSB sites. The authors should examine if MRE11 and 53BP1 localization to DSB sites is compromised after IR in GFI1-null cells.

We have performed immunofluorescence experiments to measure the formation of MRE11 and 53BP1 foci following irradiation. We find that there is no difference in foci formation for either of these 2 proteins between GFI1 WT and GFI1 KO cells (See supplementary Figure 9 a, b and text page 11, lines 8-12).

These results are consistent with those reported by Yu et al. (Cell Research 2012, reference 19) showing that a non-methylatable MRE11 R/K mutant protein could localize to sites of DNA damage (Figure 4 of their paper) but had deficient exonuclease activity (Figure 7). Similarly, Boisvert et al. (Cell Cycle 2005, reference 22) showed that mutating the arginine residues within the GAR motif of 53BP1 did not prevent its localization to sites of DNA damage (Figure 6 in their paper) although it did interfere with DNA binding itself (Figure 4). Altogether this suggests that GFI1 regulates, through their methylation, the activity of MRE11 and 53BP1 at DNA damage sites but not their recruitment to those sites.

4. Loss of GFI1 decreases the methylation of the DDR substrates MRE11 and 53BP1 by abrogating their interaction with PRMT1, suggesting GFI1 may mediate the interaction between PRMT1 and its DDR substrates. Does GFI1 directly interact with PRMT1? Does GFI1 affect PRMT1 activity? In vitro methylation assays of PRMT1 with DDR proteins in the presence or absence of GFI1 should be done. Does GFI1 affect the expression or stability of PRMT1, MRE11, and 53 BP1? Based on the Western blots in Figure 5, it appears that loss of GFI1 decreases the expression of MRE11 and PRMT1.

We carried out vitro methylation experiments of the MRE11-GAR motif by PRMT1 with increasing amounts of GFI1 protein. We observed that addition of GFI1 to the reaction did not affect the efficiency of methylation of the substrate by PRMT1 (See supplementary Figure 6 and text page 10, lines 28-30). This suggests that while GFI1 mediates the interaction of PRMT1 with its substrates in vivo, GFI1 does not affect the catalytic activity of PRMT1 itself.

Regarding the second part of the comment, we have quantified all input blots for PRMT1, MRE11 and 53BP1 from experiments comparing cells expressing different levels of GFI1 (Thymocytes, Jurkat and SupT1). These aggregated results, shown in supplementary figure 12, show that although a small amount of variation can be seen in specific panels, there is no significant change in the level of these proteins.

5. The same group reported that GFI1 promotes leukemia cell survival by restraining p53-mediated transcription activation of apoptotic genes after IR. Does DNA repair defect seen in GFI1-null cells dependent upon p53? The author should knockdown p53 expression and compare DNA repair in wild type and GFI1-null cells. This would help define a non-transcriptional role of GFI1 in DNA repair.

To address this comment, we performed a Comet assay comparing the DNA repair efficiency of cells extracted from p53 KO GFI1 WT and p53 KO GFI1 KO mice. We observed a deficiency in DNA repair in the p53 KO GFI1 KO cells compared to the p53 KO GFI1 WT cells similar to that observed in a p53 WT context (See supplementary Figure 1 j and text page 6 lines 18-21). This suggests that the effect of GFI1 in DNA repair is independent of p53 and of GFI1's own effect on p53.

Other concerns:

1. Standard deviation should be used rather than standard error throughout the manuscript. We have modified all graphs previously using standard errors to now report standard deviations and the Figure legends have been modified accordingly. We apologize for

instances where standard deviation had been used but the Figure legend mistakenly mentioned standard error. This has been corrected.

2. Figure 3a – Proteins should be labeled.

We have labeled the GFI1 protein and the IgG on the blot, however, we would hesitate to identify any additional proteins as it is not possible to state with certainty that any specific band that is visible on the gel corresponds necessarily to the specific protein identified in the mass spec with the according molecular weight, since the band corresponding to any identified protein could be too faint to see on the gel or overlap with that of a more abundant protein.

3. Figure 3e- Reciprocal CoIP should be done. In the text it states that only the loss of the intermediate domain (lane 3) affects the interaction between GFI1 and PRMT1 and MRE11 but it appears that the SNAG domain is also important as well as the DNA-binding domain for interaction with MRE11. Blots for ATM, MRE11, and PRMT1 are missing for the input samples.

We have added input blots for PRMT1 to the Figure; the extracts had unfortunately not been blotted for ATM and MRE11 at the time of this experiment. However, inputs from identically treated cells were blotted for FLAG, MRE11, PRMT1 and Actin (See Supplementary Figure 4 c).

We have performed the reciprocal Co-IP experiments successfully with ATM and PRMT1; we were however unable to detect Flag-GFI1 proteins following MRE11 IP. The successful blots can be found in Supplementary Figure 4 a, b.

We also recognize that our co-IP experiments show a reduction of the interaction for the GFI1 mutants lacking the SNAG or DNA binding domains, although some weak interaction remains detectable. However, given that re-expression of these mutants rescues the DNA repair defect in GFI1 KO cells (See Figure 3 h), it appears that the extent of interaction observed is sufficient for GFI1 to play its role in the DNA repair process.

It is also possible that the reciprocal co-IP does not fully capture the interaction between the GFI1 mutants and the proteins of interest due to differences in antibody affinity or other technical reasons such as steric hindrance.

4. Figure 3i – The endogenous CoIP of GFI1 with PRMT1 is not convincing in both cell lines.

We have repeated the co-IPs of GFI1 with PRMT1 in SupT1 and Jurkat cells using a different combination of antibodies (a goat antibody for the IP of GFI1 and a mouse antibody for blotting of PRMT1). This minimized the cross reaction with IgG and yielded a much clearer result. We have replaced the original results in Figure 3i.

5. Figure 5- in panel b, overexpression of GFI1 enhances MRE11 methylation. This maybe due to more loading of GFI1 overexpressed sample as indicated by more IgG signal.

Additionally, does overexpression of GFI1 enhance PRMT1-MRE11 interaction?

We have repeated the IP experiment and can now demonstrate increased ADMA of MRE11 more clearly (confirming our results). In addition, we have performed co-IP experiments between PRMT1 and MRE11 in order to show that the interaction between these two

proteins is indeed increased in GFI1 overexpressing cells (See Figure 5 b, e and f and text on page 10, lines 15-17).

6. For comparison of methylation level and the detection of coIP, the authors need to show the amount of target protein being IPed. This is a common issue for Figure 5 (a & b), Figure 6 (a & c), Supplementary Figure 4 (b, c & d), and Supplementary Figure 5 (a, b & c).

We have performed a series of immunoprecipitations for all relevant proteins (PRMT1, MRE11 and 53BP1) in each cell type used (Thymocytes, Jurkat and SupT1 cells) to confirm that the amount of protein immunoprecipitated was equal between the different cells used in the co-IP experiments and thus did not affect our interpretation of the results. For each protein and in each cell type, we found no significant differences in the amount of proteins IP-ed between cells (See supplementary Figure 8 and text page 11 lines 4-5).

7. Figure 6 c – the efficacy of MS023 treatment on cellular ADMA level should be shown. The quality of LaminB1 blot needs to be improved.

We have repeated the LaminB1 blot in Figure 6C using the original protein extracts and the Figure was changed accordingly.

We have treated the SupT1 and Jurkat cell lines with increasing concentrations of MS023 and blotted nuclear extracts for ADMA to show the efficacy of MS023 treatment on cellular ADMA. We show that there is a significant decrease of the ADMA signal starting around a 250-500nM concentration. These results are shown in supplementary Figure 10, panels a, b, d and e.

Reviewer #3 (Remarks to the Author):

Title: GFI1 is required for efficient DNA Repair by regulating PRMT1 dependent methylation of MRE11 and 53BP1

Summary: The authors present interesting data that implicate GFI1 in the repair of DNA double strand breaks in T cells. Upon the induction of DNA double-strand breaks, GFI1 stimulates the PRMT dependent methylation of MRE11 and 53BP1, thereby promoting repair by homologous recombination. The paper is extremely well written and if the authors can address my concerns, please see the following section, I see no reason why the paper cannot be accepted.

Major concerns:

1) The use of the alkaline comet assay for showing effects on DNA double-strand break repair. I am aware that in the text the authors state that this assay is able to detect double- and single-strand breaks, and they are correct. The problem is that the contribution of double-strand breaks to comet length, compared to single-strand breaks, are negligible. The differences in comet length presented in Figures 1, 3 and 6 might be due to differences in single-strand break repair. I would propose the authors repeat one or two of the comet assays, but this time do it under neutral conditions. See whether the differences are still apparent.

We have performed a comet assay under neutral conditions in GFI1 KO and WT Jurkat cells and we observe also under these conditions that the repair of double-strand breaks specifically is impaired in the GFI1 KO cells (See supplementary Figure 1 i and text page 6 lines 16-18).

2) The use of "mean nuclear gamma-H2AX signal" as an indication of double-strand breaks. The "brightness" of a focus is determined by kinase and phosphatase activity, not by the presence of a double-strand break. Irrespective of the brightness, foci number tells you whether there is, or was a double-strand break. Please express the graphs as the number of foci per cell, or an equivalent.

We have repeated the analysis of the gamma-H2AX IF images by counting the number of foci per cell. Figure 2, panels a and b have been modified accordingly. Our observations and interpretation of the results remain the same following this change and confirm our previous result.

3) The specific methylation of MRE11 and 53BP1 are very interesting as these two proteins are the hub for repair pathway choice. Some data are presented that the methylation of MRE11 stimulates homologous recombination, but the methylation of 53BP1 had no effect on non-homologous end joining. Please expand the discussion to include possible reasons why the cell need to methylate both.

There are two aspects we which to address regarding this issue:

First: the effect of GFI1 on 53BP1 methylation suggests that we should expect the former to have an effect on NHEJ, which we did not observe. However, other results strongly suggest that GFI1 has an effect on repair pathways other than HR. Indeed, our comet assay experiments are carried out on unsynchronized cell populations with a large proportion of cells in G1 (90% for thymocytes and 80% for SupT1 cells) and yet there is a clear effect of GFI1 on repair. In addition, new FACS based γ -H2AX signalling experiments (See

supplementary Figure 2 d) show a clear effect of GFI1 on γ -H2AX signalling in G1. These results suggest that GFI1 plays a role in repair pathways other than HR, and we speculate that the plasmid-based assay used here to measure NHEJ may not have been sufficiently sensitive to detect the effect of GFI1 deficiency.

Second: It has been reported in the literature that 53BP1 plays a role in DSB repair in heterochromatin, including in HR repair of DSBs in G2, through the regulation of KAP-1 accumulation at break sites, which represents an additional way in which the effect of GFI1 on 53BP1 methylation may affect DNA repair (see references 39 and 40, Kakarougkas A, *et al*, NAR 2013, Noon AT, *et al*, Nat Cell Biol 2010).

These points are mentioned in the discussion on page 14, lines 12-23.

Minor points:

- 1) Graphs: I think it should be a decimal point and not a decimal comma. All graphs were modified to use decimal points where applicable.
- 2) Page 8, line 12. You used "in" twice. This has been corrected.
- 3) Page 10, line 17. Remove the word "phenotypes" that appear after the word "consistent". This has been corrected.

REVIEWERS' COMMENTS:

Reviewer #2 (Remarks to the Author):

The authors put significant amount of efforts to address my critiques. The overall quality of this manuscript has been greatly improved. Both Major and minor critiques have been well covered, or if not, discussed.

At this stage, the manuscript is well fit for acceptance.

Reviewer #3 (Remarks to the Author):

The authors have addressed all my concerns and thereby have further improved an already very good manuscript. I can, therefore, recommend the paper be accepted for publication.